# Remediation and upcycling of microplastics by algae with wastewater nutrient removal and bioproduction potential

Bin Long[1,2,3,4,8], Qiang Li[1,2,8], Cheng Hu[1,2,3,4], Yayun Chen [5], Yining Zeng[6], Weiwei Li[7], Sydney Pearson[1,2], Mengqiao Liu[5], Chengcheng Fei [5], Joshua S. Yuan [1,2,7] ✉ & Susie Y. Dai [1,2,3,4] ✉

Microplastics have emerged as major environmental hazards that require efficient, cost-effective, and sustainable remediation technologies. This study introduces an integrative platform for the remediation and upcycling of microplastics by algae, while synergizing with plastic upcycling, wastewater treatment, and algal production. The strategy employs a mechanism that enhances hydrophobic interactions between the cell surface and microplastics, enabling rapid aggregation and removal. The platform achieves a superior microplastic removal efficiency of 91.4% within 1 hour, with a capacity of 0.1-gram microplastic per gram of biomass. Furthermore, the study demonstrates an upcycling strategy that converts microplastics-enriched cyanobacteria into plastic composites with unique performance. This work also integrates microplastic removal with cyanobacterial bioproduction and wastewater treatment, offering an approach that synergizes remediation with these value-added processes. Ultimately, this platform provides a viable and sustainable pathway to address microplastic pollution by creating value through plastic upcycling, wastewater nutrient removal, and $CO_2$-based bioproduction.

The 300 million tons of non-degradable petrochemical plastics disposed per year have presented significant environmental challenges[1]. Besides low recyclability, the decomposed microplastics and nanoplastics pose major threats to ecosystems, with negative impacts on a broad spectrum of living organisms, including microorganisms, plants, animals, and human beings[2–4]. In particular, a recent remarkable study shows that patients with microplastics present carotid artery plaque have a higher risk of composite outcomes of myocardial infarction, stroke, or death[5]. Recent studies indicated that microplastics and nanoplastics can accumulate up to 0.5% of brain biomass and could be

associated with a series of health issues from dementia to Alzheimer's[6]. Conventional microplastics remediation methods involve filtration and flocculation, which are not only costly, but also subject to clogging and water chemistry variations to impact the effectiveness. The current state-of-the-art microplastic remediation still primarily focuses on designing different systems to remove the microplastics from the aqueous phase[7–9]. For example, one study developed core-shell superparamagnetic nanoparticles that attract and glue various microplastics into larger agglomerates, which can then be removed from water with an external magnetic field[9]. Another study employs an

[1]Department of Plant Pathology and Microbiology, Texas A&M University, College Station, TX, USA. [2]Synthetic and Systems Biology Innovation Hub (SSBiH), Texas A&M University, College Station, TX, USA. [3]Department of Chemical and Biomedical Engineering, University of Missouri, Columbia, MO, USA. [4]Christopher S. Bond Life Science Center, University of Missouri, Columbia, MO, USA. [5]Department of Agricultural Economics, Texas A&M University, College Station, TX, USA. [6]Renewable Resources and Enabling Sciences Center, National Renewable Energy Laboratory, Golden, CO, USA. [7]Department of Energy, Environmental, and Chemical Engineering, Washington University in St. Louis, St. Louis, MO, USA. [8]These authors contributed equally: Bin Long, Qiang Li. ✉e-mail: joshua.yuan@wustl.edu; sydai@missouri.edu

interfacial solar evaporation platform to simultaneously produce clean water and significantly improve microplastic removal from the source[10]. While removing microplastics is important, such a strategy leaves a waste stream that needs storage or further processing of the spent media[11]. Even though the current wastewater treatment systems showed varied capacities in microplastic removal[12], the removal capacity decreases when the microplastics sizes get smaller[12]. It is therefore critical to advance economic, sustainable, and effective microplastic remediation technologies that could integrate with wastewater treatment and take considerations of end-of-life plastics' fate.

To address the challenges of cost and secondary waste, an ideal solution would integrate microplastic removal with a valorization strategy[13]. This reframes microplastics from an environmental hazard into a potential carbon-rich feedstock for upcycling, a key step toward a circular economy[14]. However, a significant technological barrier remains: the high cost and inefficiency of current separation methods make it unfeasible to generate a microplastic stream for conventional recycling[12]. A paradigm shift is needed, moving from simple removal to a synergistic system that can both capture microplastics and simultaneously generate value. Here, biological systems offer unique opportunities. If an organism could not only sequester microplastics but also produce a valuable co-product, such as biomass or bioplastics[15–17], the entire process might become economically viable. This approach would create a direct downstream pathway for the captured plastics, embedding them within the biomass for co-processing and eliminating the risk of re-release into the environment[18,19]. While algal biomass has been explored for bioplastics production[15–17], no current research has proposed synergizing microplastic capture with the simultaneous manufacturing of a valuable algal biomass, a strategy that could finally overcome the economic and logistical hurdles of microplastic remediation.

Among different microplastic remediation technologies, bioremediation is more environmentally friendly because microorganisms are renewable, easy to grow, and the method is compatible with various downstream processes. Conventional microbial biodegradation methods will depend on the type of plastics to be remediated and have limitations when applied to the environmental remediation where a variety of different types of microplastics exist[20]. An alternative mechanism involves cell surface adhesion, where recent studies have found a strong positive correlation between a microbe's extracellular polymeric substances (EPS) production and its microplastic removal capacity[21–23]. For example, *Gloeocapsa* sp., which produces high levels of EPS, demonstrated excellent microplastic removal capacity, while poor producers such as *Microcystis panniformis* and *Synechococcus elongatus* PCC 7942 showed minimal MP removal[21,23]. Furthermore, modulating *Pseudomonas aeruginosa* EPS production has successfully achieved 'capture and release' of microplastics[24,25]. Using microbial EPS to remove microplastics is largely dependent on the species and their capacities vary significantly[22,23]. Moreover, the EPS-based microplastic removal is relatively slow and inefficient (Table 1)[21,22,24,25]. For example, it takes *Synechococcus* sp. PCC 7002 over six hours to achieve a modest removal rate (around 80%) of microplastics by the high EPS-producing strain[21]. Furthermore, the utilization of microbial microplastics remediation will always have to take into consideration of toxicity, as the previously mentioned strains like *Microcystis panniformis* are also notorious for producing microcystin to endanger human, livestock and wildlife health[26].

Despite the limitations, microbial microplastics removal could achieve economic and sustainable process if the organism can produce useful bioproducts, remediate other contaminants, and achieve high efficiency of microplastic removal. In this regard, cyanobacteria have the potential to synergize the excess nutrient utilization. In a previous work, it was shown that engineering a hydrophobic cell surface can promote cyanobacterial self-aggregation, thus facilitating the low-cost recovery of the biomass at a high efficiency[27]. The engineering

**Table 1 | Summary of microorganism-based microplastic capture**

| Species | Mechanism | Design | Microplastic type and size | Temperature | Efficiency | Ref. |
|---|---|---|---|---|---|---|
| M. panniformis | EPS | N.A. | PS (<106 μm) and PMMA (<250 μm) | 20 ± 1 °C | N.A. | 23 |
| Scenedesmus sp. | | | | | | |
| Tetraselmis sp. | | | | | | |
| Gloeocapsa sp. | | | | | | |
| Cyanothece sp. | EPS | N.A. | PS (0.1 and 10 μm) | 25 ± 1 °C | N.A. | 22 |
| Synechococcus PCC 7942 | EPS | N.A. | PS (0.1 and 10 μm) | 30 °C | ~18% in 6.5 h[#] | 21 |
| Synechococcus PCC 7002 | | | | | ~ 82% in 6.5 h[#] | |
| P. aeruginosa | EPS | Knockout of wspF & inducible expression of yhiH | PET, PMMA, nylon, PVC (<106 μm), Microplastics from seawater (106–300 μm) | 25 °C or 30 °C | >90% in 24 h[#] | 24 |
| P. aeruginosa | EPS | Expression of typsin & laboratory evolution | PS, PET, PMMA (<106 μm), Microplastics from seawater (106–300 μm) | 37 °C | N.A. | 25 |
| Synechococcus UTEX 2973 | Hydrophobicity | Expression of limonene synthase | PS (<5 μm), PET (<300 μm), PE (35 μm), Microplastics from surface water and wastewater. | 37 °C and room temperature | 91.4% in 1 h | This study |

N.A. not appliable or not indicated in the original paper, EPS extracellular polymeric substances, PS polystyrene, PMMA polymethyl methacrylate, PET polyethylene terephthalate, PVC polyvinyl chloride, PE polyethylene.
[#] Estimated from figures.

design was achieved through computational modeling-guided synthetic biology design of high limonene production, and the base strain *Synechococcus elongatus* UTEX 2973 (UTEX 2973) is known for high productivity and absence of microcystin producing genes[27–29]. Such hydrophobic-mediated cell-to-cell interactions and self-aggregation inspired us to explore whether hydrophobic effects could also drive interactions between hydrophobic cells and microplastics, given that most plastics are highly hydrophobic. This approach could not only offer an efficient and sustainable method but also introduce a distinct mechanism for microplastic remediation.

This work presents a conceptual design of RUMBA, a technology for the remediation and upcycling of microplastics by algae, which synergizes microplastic removal, wastewater nutrient utilization, and plastics upcycling. The synthetic biology design of the limonene-producing strain facilitates self-aggregation of the microplastics-enriched biomass and removal from the liquid phase, empowered by the hydrophobic interactions within the cyanobacterial population and between the cells and microplastics. Critically, this study demonstrates two possible sustainable and value-added processes to utilize the algal biomass and recovered microplastics. First, the engineered cyanobacteria have achieved efficient utilization of nutrients from wastewater, with a unique potential to integrate with wastewater treatment. Second, the recovered microplastics can be upcycled into bioplastic films with algal biomass as biofillers. Overall, RUMBA presents an approach to synergize microplastic remediation, upcycling, and nutrient removal by utilizing modified cell surface features and the nutrient up-taking capacity of engineered cyanobacteria. The platform has significant sustainability and economic benefits.

## Results

### Highly efficient microplastic removal by hydrophobic cyanobacteria cell (HCC)

Microplastics have been characterized as hydrophobic with large surface areas[30], which laid down the hypothesis for RUMBA to achieve removal design by enhancing hydrophobic interaction between microplastics and the engineering cyanobacterial strain. Our previous study has shown that hydrophobic cyanobacteria cells (HCC) can self-aggregate and achieve auto-sedimentation[27]. The HCC was engineered through synthetic biology design of UTEX 2973 for limonene production[27]. The produced limonene was found to be secreted and enriched on the cell surface before volatilizing, thereby increasing cell hydrophobicity[27]. The unique smooth cell surface of UTEX 2973, resulting from defects in pilus biogenesis, exposes the hydrophobic surface, which enables interactions that promote cell aggregation (Fig. 1a)[27]. Indeed, cell aggregation and self-sedimentation were observed in HCC cells, but not in wild-type (WT) cells (Fig. 1b–f). This behavior is linked to limonene production in the HCC samples (Supplementary Fig. 1a–d), which increases cell surface hydrophobicity. A subsequent BATH assay confirmed this enhanced hydrophobicity, demonstrating that a larger portion of HCC cells attached to a hydrophobic hexadecane layer compared to WT cells (Fig. 1g, and Supplementary Fig. 1e). These results corroborate with our previous results[27] and zeta potential data Fig. 1h. Thus, it is reasonable to hypothesize that the hydrophobic cell surface could effectively interact with the hydrophobic portion of the plastic surface, leading to cell-microplastic aggregation, co-sedimentation, and subsequent microplastic removal (Fig. 1a).

To test this hypothesis, the effectiveness of HCC in interacting with and removing polystyrene (PS) microplastics was evaluated. HCC demonstrated remarkable PS removal capacity across all tested sizes, as characterized by enhanced PS-HCC co-sedimentation (Fig. 1b). When cyanobacterial cells were mixed with 200 nm PS microplastics, substantial sedimentation was observed in the HCC samples but not in the WT samples after settling for one hour (Fig. 1b). Particularly, the sedimentation process was accelerated and enhanced in samples with

higher PS concentration, indicating potential interactions between HCC and microplastics (Fig. 1b and Supplementary Fig. 2). To better understand the interaction and evaluate microplastic removal efficiency, suspension and sediment analyses were conducted to estimate their compositions. The suspension turbidities shown in Supplementary Fig. 1b estimate the total solid contents, including both cyanobacterial cells and PS microplastics in the suspension. The results highlight significantly ($p < 0.01$) lower turbidities in HCC samples compared to WT samples across various PS concentrations (Fig. 1b). Additionally, the chlorophyll fluorescence in the suspension of HCC samples was significantly lower ($p < 0.01$) than in WT samples at all PS concentrations, indicating a decreased presence of cyanobacterial cells in HCC samples than in WT samples in the presence of microplastics (Supplementary Fig. 1c). The results thus clearly show that HCC interacts with microplastics to achieve co-sedimentation for removal.

To further evaluate the PS contents in the suspensions, low-speed (i.e., 800 × g for 200 nm PS microplastics and 300 × g for 500 and 800 nm PS microplastics) centrifugation was employed to separate cyanobacterial cells and PS microplastics. Since the size and density of microplastics (1.005 g/mL) and cyanobacteria cells are different, differential centrifugation thus pelletizes the cell first and leaves most microplastics in the suspension at the low centrifugation speed. The centrifugation step was able to pelletize >95% of cells and retain 82.3% to 97.7% PS microplastics in suspension, depending on their sizes (Supplementary Fig. 3), which legitimizes the effectiveness of the differential centrifugation. The turbidity of the centrifuged suspension was then used to estimate PS concentrations. As expected, the turbidities of centrifuged suspension were significantly lower ($p < 0.05$) in the HCC samples than in those of the WT samples across all tested PS concentrations, suggesting HCC removes a significant amount of PS microplastics from the solution phase (Fig. 1c). Meanwhile, visibly fewer pellets were observed in HCC samples compared to WT samples after centrifugation (Supplementary Fig. 4). The results were observed from PS microplastics across different sizes (Supplementary Fig. 5). Furthermore, the long-term microplastic removal trial at 0.5 L demonstrated that sedimentation remained effective throughout a continuous 19-day period (Supplementary Fig. 1f). Collectively, the results suggest that the HCC interacts with PS to achieve highly efficient removal of microplastics.

The efficiency of microplastic removal by HCC was evaluated through three steps. First, the percentages of dry weight in suspensions and sediments for both WT and HCC samples were analyzed. To account for the absence of sediments in WT samples, we consistently took the bottom 5 ml of samples as sediments for both WT and HCC. While this sampling strategy likely led to an overestimation of WT (and WT-PS)'s microplastic removal capacity, its impact on HCC samples should be limited due to the relatively low cell concentrations in the suspensions. The dry weight measurements suggest that sediments account for 11.9% and 82.5% of the total mass for WT and HCC samples in the absence of PS, respectively (Fig. 1d). In the presence of PS, the sediment proportion for WT samples showed a slight increase to 12.7%, while HCC samples exhibited a significant increase ($p < 0.05$) to 90.3% (Fig. 1d). Second, thermogravimetric analysis (TGA) was applied to further analyze the suspension compositions. The results reveal that microplastic dry weight constituted 65.3% and 37.8% of the total suspension mass for WT and HCC suspensions, respectively (Fig. 1e). Third, the removal rate was calculated based on the results from the first and second steps using Eqs. 1–3 in the Materials and Methods section. The calculation highlighted that 91.4% of PS microplastics were removed by HCC within 1 hour. Such efficiency was superior to previously developed EPS-dependent microplastic bioremediation methods (Table 1).

Furthermore, the microplastic removal capacity was evaluated by measuring the cell biomass required to remove 5 mg of 200 nm PS

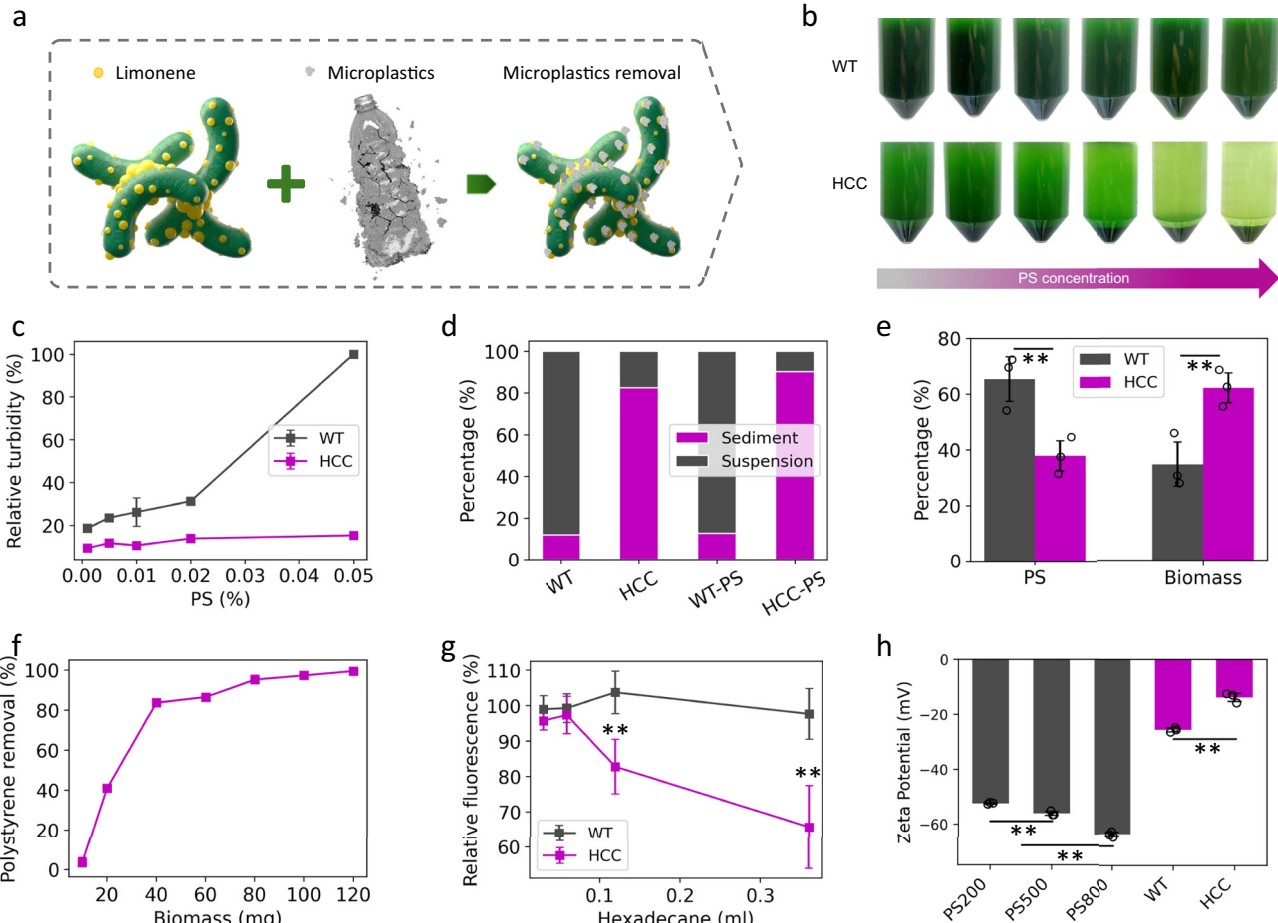

**Fig. 1 | Microplastic removal by hydrophobic cyanobacterial cells (HCC).**
**a** Illustration of the hydrophobicity-mediated cell-limonene-microplastic interaction. **b** Gradient concentrations of 200 nm PS microplastics were mixed with (wildtype) WT and HCC cells at final concentrations of 0%, 0.001%, 0.005%, 0.01%, 0.02%, and 0.05% (from left to right). Sedimentation was observed in HCC-PS samples, but not in WT-PS samples. Moreover, the sedimentation increases as the PS concentrations increases. Further analysis of the suspension with low-speed centrifugation, which separated microplastics from cyanobacterial cells, demonstrates significantly ($p < 0.05$ except at 0.01% PS, where $p = 0.14$) lower polystyrene abundance (**c**). The reading of the WT sample containing 0.05% PS microplastics was normalized to 100% to streamline the comparison. **d** Sediment fractions in HCC samples were found to be significantly ($p < 0.01$) higher than WT samples.
**e** Significantly higher PS fractions ($p = 0.016$) were observed in WT-PS suspensions (65.3%) compared to HCC-PS suspensions (37.8%). **f** The observation that 40 mg of

HCC were able to remove 83.7% of 5 mg of PS microplastics indicates a removal capacity of ~0.1 g of PS microplastics per gram of biomass. **g** Hexadecane was added in varying volumes to cyanobacterial cultures ($OD_{730} = 0.2$), followed by vortexing. Increased cell hydrophobicity resulted in greater adherence to the hexadecane layer, thereby reducing the OD of the aqueous phase. With increasing hexadecane volumes, the aqueous phase OD of HCC samples was significantly lower than that of WT samples ($p = 0.0007$ for 0.12 ml and 0.0016 for 0.36 ml), indicating enhanced hydrophobicity in HCC cells. **h** Zeta potential of PS microplastics, WT cells, and HCC cells. The zeta potential of PS microplastics significantly decreases ($p < 0.01$) with increasing diameters. Moreover, the zeta potential of HCC cells (−13.91 mV) is significantly higher ($p = 0.0005$) than that of WT cells (−25.78 mV). Data are presented as mean values ± standard deviations ($n = 3$ independent samples). ** indicates $p < 0.01$. Two-tailed Student's t-test was used. Source data are provided as a Source Data file.

microplastics, an amount that is measurable and also within the removal capacity by HCC cells. Figure 1f shows that 40 mg of HCC are sufficient for removing 83.7% of 5 mg of PS microplastics, leading to an approximate removal capacity of 0.1 gram of microplastics per gram (dry weight) of HCC. To our knowledge, most previously published bioremediation systems have not quantified their removal capacities[22–25,31]. Although the removal capacities can vary depending on the type and size of microplastics as well as the environmental conditions, it is an important factor for evaluating the potential of these systems.

**Mechanistic study of the interaction between microplastic and cyanobacteria**

Comprehensive microscopy analyses were carried out to understand the mechanisms for the interactions between microplastics and HCC. The transmission electron microscopy (TEM) analysis clearly shows

the substantial interaction between PS microplastics and HCC (Fig. 2a). Fewer PS microplastics are attached to the WT cells, and such attachment is randomly distributed (Fig. 2a). In contrast, significantly more PS microplastics have attached to HCC cells, with PS enriched at the HCC cell intersections, where limonene is expected to be present to cause cell aggregations due to its hydrophobic chemical structure (Fig. 2a)[27]. Similar patterns were also observed by SEM (Supplementary Fig. 6). These observations suggest that limonene might have caused the hydrophobic cell surface and subsequent interaction with microplastics.

The molecular mechanisms were further explored by Stimulated Raman Scattering microscopy (SRS) analysis, as stimulated Raman scattering has a superior capacity to characterize chemical composition in a non-disruptive way. The platform can identify the chemical interaction between limonene and PS microplastics[27]. The results further support the involvement of limonene in the removal of PS

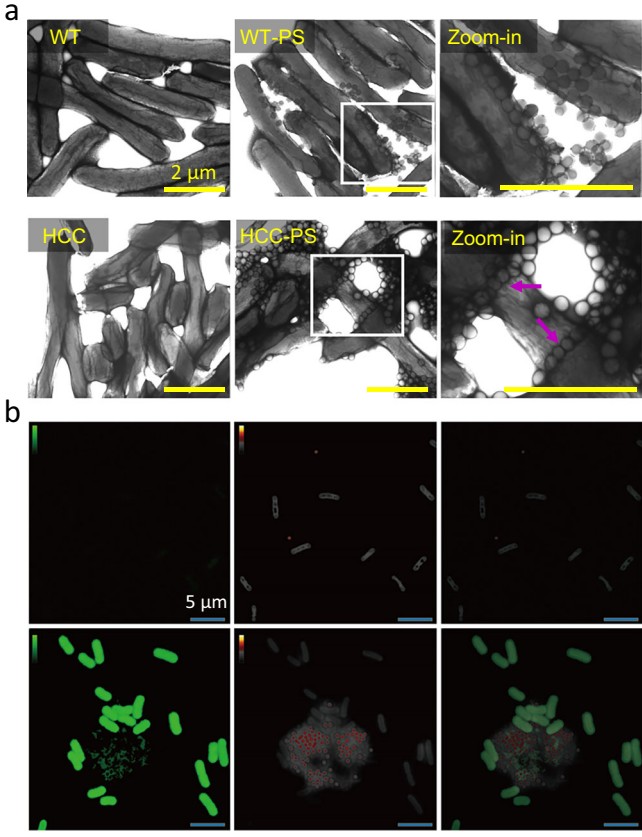

**Fig. 2 | Mechanism of the microplastic removal by HCC.** To verify that limonene is the primary driving force and mediator of microplastic capture by HCC, we investigated polystyrene-cell interactions using TEM (**a**) and SRS (**b**). The TEM images demonstrate that polystyrene (PS) randomly attached to WT cells, whereas PS microplastics were enriched at intersections of HCC, where limonene is expected to be present to drive cell aggregation (**a**). The areas within the white boxes in the WT-PS and HCC-PS samples were further magnified and displayed in the third columns to highlight the detailed interactions (Zoom-in). These results strongly support the active microplastic capture ability of HCC. The SRS images (**b**) verified the presence of limonene signals (green) in HCC-polystyrene samples and polystyrene signals (red) in both WT-PS (top row) and HCC-PS (bottom row) samples. Merging the two signals indicates that the interaction between HCC and polystyrene is driven by hydrophobic interactions between limonene and PS microplastics (third column in **b**). The experiments were independently repeated three times with similar results.

microplastics. The results highlighted the strong intracellular and extracellular limonene signals in HCC cells (Fig. 2b, green, first column), while no such signal was present in the WT samples. Meanwhile, PS microplastics were visualized at 2900 cm$^{-1}$ CH$_2$ frequency. A weak signal was found in cyanobacterial cells and was considered as background due to the presence of CH$_2$ bonds in cyanobacteria (Fig. 2b, second column). In contrast, a stronger CH$_2$ signal was observed in PS microplastics (Fig. 2b, red, second column). Overlaying the two signals revealed direct interactions between limonene and PS microplastics (Fig. 2b, third column), indicating that limonene mediates PS-HCC aggregation and ultimately leads to co-sedimentation and microplastic removal. Furthermore, measurements of extracellular polysaccharides, the major components of EPS, revealed that WT cells produced a higher level of extracellular polysaccharides compared to HCC cells (Supplementary Fig. 1g). This finding indicates that the observed cell-microplastic interaction in HCC is likely independent of EPS production, further supporting the role of limonene-induced surface hydrophobicity.

Additionally, the disruption of hydrophobic interactions impairs the microplastic removal by HCC was studied. A surfactant, Tween 20, was introduced to modify the hydrophobic surfaces of cyanobacterial cells and PS microplastics (Supplementary Fig. 7a). With the addition of Tween 20, we have observed increased suspension turbidity in the HCC samples. The results indicate that disrupting hydrophobic interactions significantly suppress cell-cell and cell-microplastic aggregation and disable aggregation-based sedimentation ($p < 0.01$, Supplementary Fig. 7b). After the Tween 20 treatment, the low-speed centrifugation was employed to separate the cells and PS microplastics in the suspension to quantify the PS content, as shown in Supplementary Fig. 7c, the turbidity of the centrifuged suspension significantly ($p < 0.01$) increased by 48.3% and 1020.9% in WT and HCC samples, respectively, when Tween 20 was added. These results highlight the crucial role of hydrophobic interactions in microplastic removal by HCC.

Furthermore, the zeta potential measurements were performed to further elucidate the interaction mechanisms between HCC and microplastics. The PS microplastics exhibited consistently negative zeta potentials: $-52.42 \pm -0.26$ mV, $-56.16 \pm -0.73$ mV, and $-63.88 \pm -0.69$ mV for 200 nm, 500 nm, and 800 nm sizes, respectively (Fig. 1h). In contrast, HCC cells demonstrated a significantly ($p < 0.01$) lower zeta potential of $-13.91 \pm -1.47$ mV, compared to $-25.78 \pm -0.69$ mV for WT cells (Fig. 1h). This reduction in the negative surface charge on HCC cells can be attributed to their increased surface hydrophobicity, which likely lessens the adsorption of polar molecules and charged ions. This, in turn, leads to a reduced negative charge in the stern layer of HCC cells and a diminished overall zeta potential. Taken together, these results highlight the dual roles of increased cell hydrophobicity in mediating HCC-PS interactions: it weakens electrostatic repulsion by decreasing the cell's negative charge, while simultaneously enhancing cell-to-microplastic attraction through increased hydrophobic interactions.

## HCC interacts with a broad spectrum of microplastics

Based on the same mechanism to capture PS microplastics, we further examined the ability of HCC to remove other types of microplastics, such as polyethylene terephthalate (PET) and polyethylene (PE). Due to the large microplastic sizes (< 300 microns for PET and 32–38 microns for PE), the sedimentation assay used for PS microplastic removal quantification is not suitable for the PET-HCC and PE-HCC interactions. As an alternative, the attachment of microplastics by HCC is evaluated under microscopy. As shown in Fig. 3a–l, cyanobacterial cells show red color with chlorophyll fluorescence, while the autofluorescence of PET and PE shows cyan. Cell aggregates are observed in the samples with HCC (Fig. 3d, j) but not in WT samples (Fig. 3a, g). Overlaying the chlorophyll fluorescence of cyanobacteria with the auto-fluorescence of PET and PE, cell-microplastic interactions are observed in HCC samples (Fig. 3f, l). In contrast, there are barely any interactions in the WT samples (Fig. 3c, i). In particular, zoom-in images clearly show the attachment of cyanobacterial cells/aggregates on PET (Fig. 3m) and PE (Fig. 3n) microplastics. These results validate the interactions between the hydrophobic cells and microplastics, suggesting that HCC can potentially be used to remediate a wide spectrum of microplastics.

## HCC removes microplastics at environmentally relevant conditions for water treatment

Given the widespread presence of microplastics in almost all natural water systems[32], the performance of the hydrophobic-mediated microplastic removal under environmentally relevant conditions was further evaluated, using natural surface water and wastewater samples. To do so, water samples from different locations were collected, including surface water samples from a lake in the research park (30.602603, −96.360686) on the Texas A&M University campus, as

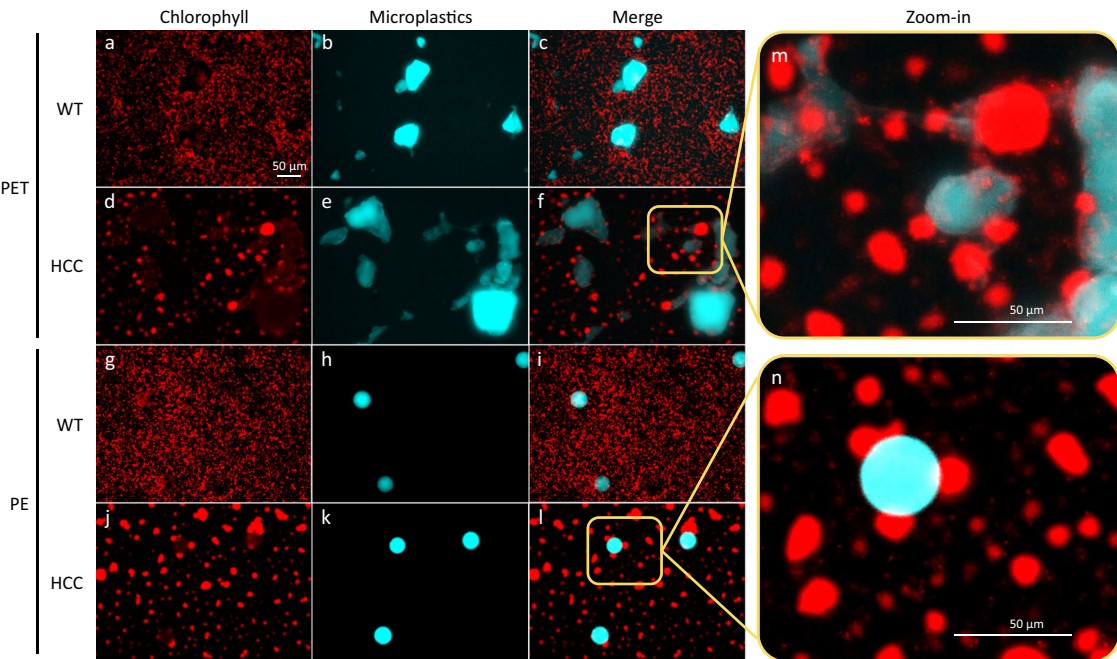

**Fig. 3 | Polyethylene terephthalate (PET) and polyethylene (PE) captured by HCC.** The capacity of HCC to capture PET (**d**–**f**) and PE (**j**–**l**) was evaluated using fluorescence microscopy, with WT cells (**a**–**c** for PET and **g**–**i** for PE) as controls. The cyanobacterial cells were visualized by examining chlorophyll fluorescence (Ex 546/10, Em 585/40), while the autofluorescence of microplastics was visualized using the DAPI filter (Ex 350/50, Em 460/50). The merged images (**f** for PET and **l** for PE) and zoom-in images (**m**, **n**) clearly demonstrate interactions between HCC and microplastics, whereas no such interactions were observed in WT samples (**c**, **i**). These results support HCC's capacity in capturing microplastics. The experiments were independently repeated three times with similar results.

well as wastewater samples from the Texas A&M wastewater treatment plant (30.564525, −96.370141).

To evaluate the efficacy of hydrophobicity-mediated aggregation and sedimentation in microplastic removal, PS microplastics of different sizes were spiked (i.e., diameters of 200 nm, 500 nm, and 800 nm to a final concentration of 0.02%) into collected water samples. The effectiveness of microplastic removal by strain HCC was then assessed by measuring the turbidity of microplastics (separated by low-speed centrifugation) in suspension, with WT cells serving as baseline controls. Specifically, the microplastic turbidity in WT suspensions was set to 100% to account for errors in sample preparation (e.g., centrifugation). Compared to WT samples, the turbidity in HCC suspensions significantly ($p < 0.01$) decreased by approximately 90% in both wastewater and surface water conditions when 500 nm and 800 nm microplastics were introduced (Fig. 4a). Interestingly, a significant ($p < 0.01$) but slightly lower reduction, approximately 80% decrease, was observed in both water samples when 200 nm PS microplastics were introduced (Fig. 4a). This lesser reduction was primarily attributed to the unexpected partial sedimentation that occurred in the WT sample with 200 nm PS microplastics.

Additionally, the interaction of HCC cells with PS, PET, and PE in surface water and wastewater environments was investigated. Clear interactions were observed between HCC and microplastics in both surface water and wastewater environments (Supplementary Fig. 8). In contrast, no significant interactions were observed between WT cells and microplastics (Supplementary Fig. 8). Overall, the results clearly demonstrate the capability of hydrophobicity-based microplastic removal under environmentally relevant conditions.

## HCC removes environmental microplastics from the surface water and wastewater

To further assess the capability of HCC in removing environmental microplastics, the environmental microplastics were enriched by isolating them from water samples collected at the aforementioned

locations and spiking them back into a small volume of their respective water samples. For instance, microplastics isolated from approximately 200 L wastewater were spiked back into 1 ml wastewater. The Nile red was used to stain microplastics before the microplastics spiking into the environmental samples. Subsequently, WT and HCC cells were added to the resulting samples, and the interaction between the cells and microplastics were observed under a microscope. Figure 4b demonstrates interactions between HCC and microplastics, while no noticeable interactions are observed in WT samples. The HCC cells clearly aggregate on certain portions of the surface of microplastics, indicating that environmental microplastics possess hydrophobic regions that allow HCC binding, regardless of their charges on other areas. These findings further validate the effectiveness of microplastic removal by HCC, highlighting its potential as an efficient tool for downstream environmental microplastic upcycling.

## Integration of microplastics removal with wastewater treatment and cyanobacterial bioproduction

The capacity of hydrophobicity-driven microplastic removal by cyanobacteria in environmentally relevant conditions opens an avenues for integrative treatment of both microplastics and wastewater simultaneously. Cyanobacteria have recently emerged as a promising approach for wastewater treatment to remove excessive nutrients[33,34]. Furthermore, cyanobacterial bioproduction has significant potential in converting $CO_2$ into value-added products, mitigating climate change, and addressing renewable product needs[35–37]. Cyanobacterial cultivation for $CO_2$ utilization, biofuels, and bioproducts requires a large quantity of water. The integration of microplastic removal with wastewater treatment and cyanobacterial cultivation using RUMBA thus could create a path to valorize the process and treat a large quantity of wastewater.

Specifically, the impact of microplastics on cyanobacterial growth was assessed. Growth assays were conducted using microplastics of varying types and sizes. No adverse effects were found on

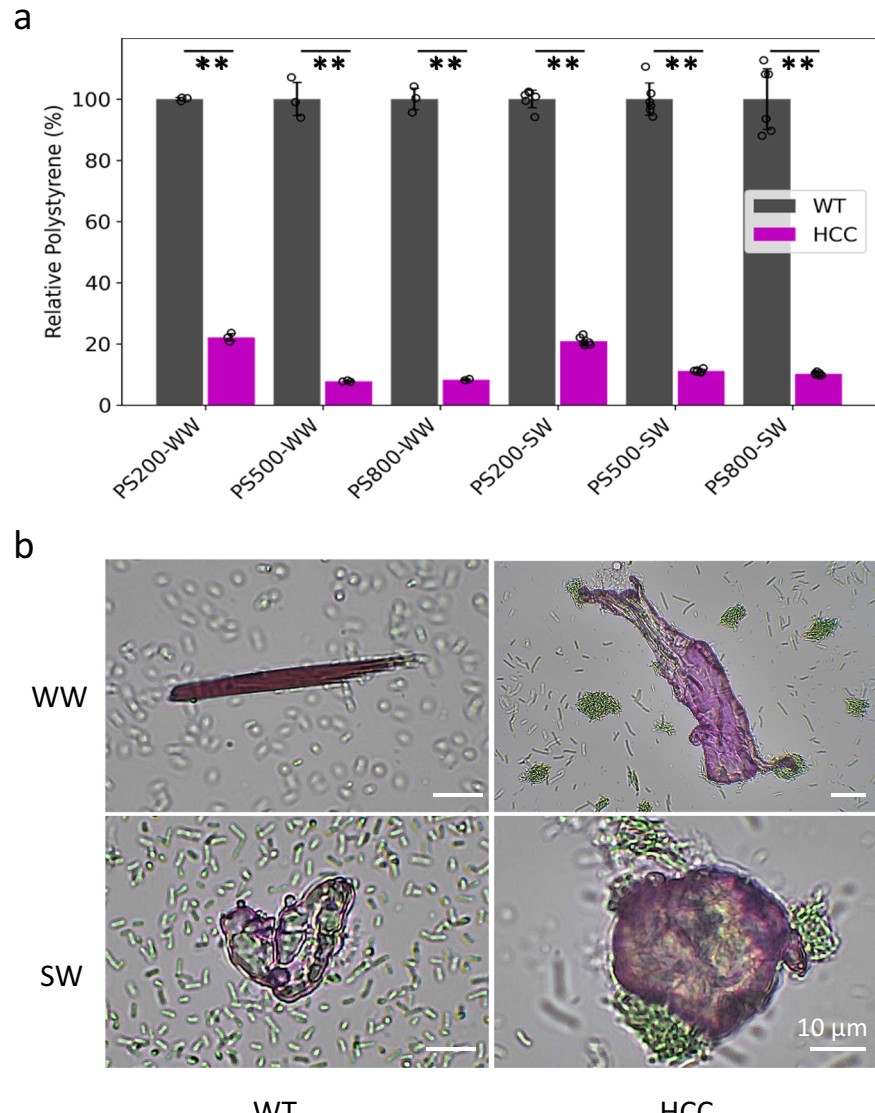

**Fig. 4 | Assessment of microplastics removal in environmentally relevant conditions. a** PS microplastics with diameters of 200 nm (PS200), 500 nm (PS500), and 800 nm (PS800) were spiked into surface water (SW) and wastewater (WW) samples. The microplastic removal capacity of HCC in these environments was evaluated by measuring the turbidity of microplastics in suspension, which were separated by low-speed centrifugation. WT cells served as baseline controls. The turbidity in the suspensions of HCC samples was substantially lower compared to WT samples, indicating effective microplastic removal under these conditions.

**b** Microscopy observations of the interactions between HCC and microplastics collected from surface water and wastewater samples. Noticeable interactions were observed in the HCC samples but not in the WT samples, suggesting the capacity of HCC in removing environmental microplastics. Data are presented as mean values ±standard deviations ($n = 3$ independent samples for WW and 6 independent samples for SW). ** indicates $p < 0.01$. Two-tailed Student's t-test was used. Source data are provided as a Source Data file.

cyanobacterial growth when exposed to low concentrations ( < 10 mg/ L) of 200 nm and 800 nm PS microplastics (Supplementary Fig. 9), which is consistent with previous findings[38]. However, increasing concentrations of PS microplastics resulted in growth inhibition, with larger microplastics exhibiting less inhibition (Supplementary Fig. 9). Specifically, significant growth inhibition ($p < 0.05$) was observed at a concentration of 0.05 g/L for 200 nm PS microplastics, whereas for 800 nm PS microplastics, significant growth inhibition was not observed until a concentration of 0.5 g/L (Supplementary Fig. 9). Moreover, PET microplastics had marginal impacts on growth, even at concentrations up to 5 g/L (Supplementary Fig. 9), likely due to their larger particle sizes ( < 300 μm). Considering the relatively low concentrations of microplastics typically found in the environment compared to those tested, we anticipate limited growth inhibition from

microplastics when combined with cyanobacterial cultivation for bioproduction.

Next, the HCC's capacity to reduce the nutrient contents of wastewater samples was investigated. The wastewater samples were collected from influent and effluent of the wastewater treatment plant, focusing on the concentration of nitrate, ammonia, and phosphate. Influent water has a low nitrate concentration of 2.1 mg/L and a high ammonia concentration of 6.8 mg/L. In contrast, the nitrate concentration in effluent increased to 64.7 mg/L, while the ammonia content fell below the detection limit. Such changes could result from the nitrification in the wastewater treatment process. On the other hand, the phosphate content slightly changed from 8.45 mg/L in influent to 10.22 mg/L in effluent. Over a period of 5 days, the engineered cyanobacterium strain demonstrated the ability to remove

**Table 2 | Nitrate removal, microplastic removal, and biomass production in the algal cultivation system that removes microplastics and nutrients**

| Treatment | Nitrate removal (%) | Microplastic removal (%) | Biomass production (g/L) |
|---|---|---|---|
| Influent- day 5 | 98.7 ± 1.0 (a) | 35.8 ± 9.1 (c) | 2.46 ± 0.28 (b) |
| Effluent- day 5 | 99.4 ± 0.2 (a) | 54.3 ± 7.9 (b) | 2.54 ± 0.28 (b) |
| Influent- day 8* | 97.6 ± 0.1 (b) | 78.5 ± 6.4 (a) | 3.80 ± 0.01 (a) |
| Effluent- day 8* | 98.6 ± 0.3 (a) | 88.6 ± 1.0 (a) | 3.50 ± 0.10 (a) |
| Influent- day 9* | 98.3 ± 0.1 (a) | 51.2 ± 1.1 (b) | 3.57 ± 0.20 (a) |
| Effluent- day 9* | 99.2 ± 0.4 (a) | 65.1 ± 12.8 (b) | 3.84 ± 0.19 (a) |

Means followed by the same letter are not significantly different. For example, for all the results with (a), they are not different from one another. For results assigned with (a) and (b) separately, the results with (a) are different from the results assigned with (b).

* Additional nutrients (equivalent to 1× BG11) were fed after day 5.

47.4% and 97.5% of nitrate from the influent and effluent, respectively, along with nearly complete removal of ammonia from the influent (Supplementary Table 1). Additionally, 34.6% and 37.8% of phosphate were removed from the influent and effluent, respectively (Supplementary Table 1). Interestingly, the addition of cyanobacterial growth media (BG11) to wastewater samples significantly enhanced nutrient removal. Ammonium and phosphate removal increased to nearly 100% in both influent and effluent samples ($p < 0.01$), while nitrate removal in the influent reached 96.7% ($p < 0.01$) (Supplementary Table 1). Nitrate removal in the effluent showed no significant changes due to its high baseline (Supplementary Table 1).

Wastewater treatment, cyanobacterial bioproduction, and microplastic removal were further integrated into a customized photobioreactor, where 0.05 g PS microplastics were spiked into 500 ml cyanobacterial media made from wastewater samples. Based on the enhanced nutrient removal capacity with the addition of BG11, the BG11 media were supplied into both influent and effluent samples. Over a period of 5 days, the engineered strain was able to remove 35.8% and 54.3% PS microplastics in the influent and effluent context, respectively (Table 2). Meanwhile, the strain removed 98.7% and 99.4% nitrate in the influent and effluent contexts, respectively (Table 2). At the same time, the strain produced 2.46 grams/L and 2.54 grams/L of biomass in influent and effluent media, respectively (Table 2). The fed-batch cyanobacterial cultivation further enhanced the microplastic removal to 78.5% in influent and 88.6% in effluent at day 8 (3 days after fed-batch) while increasing the total biomass yield to 3.80 grams/L and 3.50 grams/L (Table 2). Extending the treatment to day 9 further enhanced nutrient removal but decreased the microplastic removal capacity in both influent and effluent samples (Table 2). Samples from both day 5 and day 9 showed less optimized cell aggregation and sedimentations as compared to those of Day 8, which may result from nutrient limitation, leading to lower limonene production and reduced cell hydrophobicity. Overall, our results demonstrated the possibility of integrating microplastic remediation, waste nutrient removal, and cyanobacterial bioproduction. Depending on the wastewater composition and the bioproduction goal, we could optimize and prioritize nutrient recycling, bioproduction yield, and microplastic removal by fine-tuning the conditions.

## Upcycle microplastics as bioplastics composites
Besides wastewater treatment and bioproduction, the RUMBA process can also provide opportunities for microplastic upcycling, given that the sediments contain both cyanobacterial biomass and microplastics, both of which could be used for downstream manufacturing. For example, cyanobacterial biomass has been applied as biopolymer fillers for composite plastic films[39,40]. Thus, the removed microplastics could be upcycled together with cyanobacterial biomass to make a

plastic composite, which eliminates the need for storage and further treatment of microplastics absorbed in spent adsorbents. The upcycling process procedure is shown in Fig. 5a, b. As shown in Fig. 5c, the pure PS film is almost clear, but the composite films made from sediments show a "golden green" color (Fig. 5c), which could be due to the presence of cyanobacterial pigments in the film. The spectral analysis shows that the absorbance peaks are similar to those of chlorophyll and carotenoid in the composite films (Supplementary Fig. 10a), suggesting that these cyanobacterial pigments have been extracted and cast into the film.

The mechanical tests revealed unique features of the upcycled plastic composite. Figure 5d shows that composite films exhibit distinct mechanical properties compared to pure PS films. Both composite films have lower tensile strength, with 52.9% and 66.5% of the strength of the PS films for WT- and HCC-PS composite films, respectively (Fig. 5e). While there are no significant differences in modulus of elasticity between pure PS and WT-PS films (Supplementary Fig. 10b), elongation and toughness are significantly ($p < 0.05$) improved for the HCC-PS films compared to both pure PS and WT-PS films (Fig. 5f, g). Specifically, the elongation and toughness of the HCC-PS bioplastics were 2.3 and 2.2 times higher than that of the pure PS film, respectively (Fig. 5f, g), indicating superior properties of the upcycled bioplastics. The performance improvement could be attributed to the addition of biopolymers from cyanobacteria as fillers. Our results demonstrate a promising pathway for upcycling microplastics to enable a circular carbon economy, where recycled bioplastic composites from photosynthetic $CO_2$ fixation and microplastics could potentially serve as alternatives to petroleum-based plastic products (Fig. 5b).

## Techno-economic and Life Cycle Analysis
The techno-economic Analysis (TEA) and life cycle analysis (LCA) were carried out to evaluate the feasibility and scenario for commercial application of RUMBA, along with its emission impacts (Supplementary Tables 2–8, Supplementary Figs. 11–14, and Supplementary Data 1). LCA was performed to evaluate the environmental impact of the RUMBA process integrating with algae cultivation, microplastic removal, wastewater treatment, and bioplastic production. The functional unit, system boundary, and inventory analysis are detailed in the Supplementary Fig. 11 and Supplementary Table 2. If the system is powered by conventional energy, the total emission ranges from 19.38 to 20.67 kg $CO_2$ equivalent per 1 kg of bioplastics, with electricity consumption remaining to be the primary contributor to $CO_2$ emissions across the RUMBA-based wastewater treatment and bioplastic production life cycle (Scenarios 3 and 4 in Supplementary Fig. 12). To promote a circular economy for algae, we also consider the use of renewable energy to power the system[41], which is very feasible as the algal production can be readily coupled with solar energy since both need sunlight. Without considering residual biomass allocation, the production of 1 kg of bioplastics through upcycling results in a net utilization of 3.21 kg $CO_2$ emissions using renewable power sources. The actual industrial implementation will certainly carry out the byproducts displacement, and the scenario analysis highlights the potential for this process to achieve negative emissions (−4.50 kg $CO_2$ emissions/kg of bioplastic produced when considering residual biomass for electricity generation). This stands in contrast to other bioplastic production methods, which typically have positive $CO_2$ emissions ranging from 0.6 to 282.6 kg $CO_2$ emissions/kg of bioplastic produced[42,43]. In addition to impacts on greenhouse gas emissions, this process confers positive environmental impacts by utilizing nitrogen and phosphate from wastewater and removing microplastics[44]. The results highlighted that the synergy of microplastics removal and upcycling, algal production, and wastewater nutrient usage could achieve a significant positive environmental impact, as shown in Supplementary Data 1.

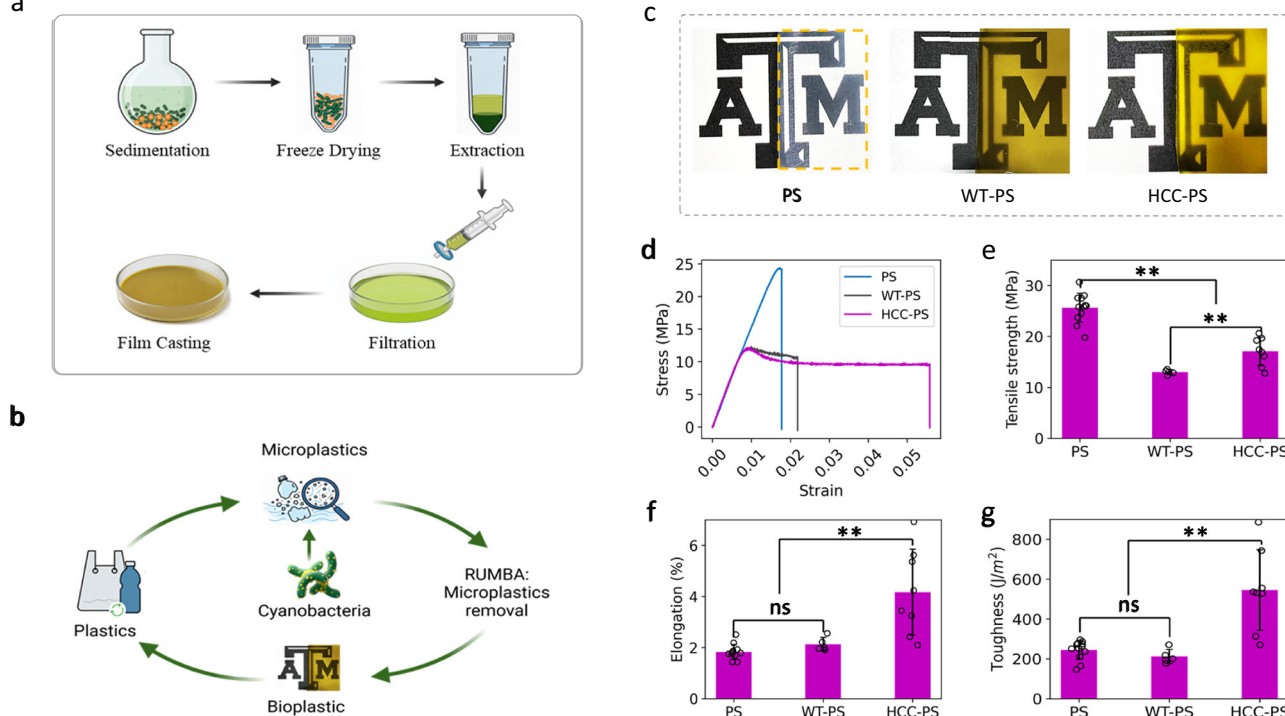

**Fig. 5 | Demonstration of microplastic upcycling and bioplastic production.**
**a** The steps of processing sediments containing polystyrene (PS) microplastics and HCC biomass yields from RUMBA to produce bioplastics. **b** The concept of the RUMBA contributes to microplastic remediation, bioplastic composite production, and the circular economy. **c** Bioplastic composite films produced from the sediments containing mixtures of PS microplastics and HCC or WT cells. A pure polystyrene film was made as benchmark control. The strain-stress curves of the films are shown in (**d**). Although the tensile strength of the bioplastic films made from WT-PS and HCC-PS were only 52.9% and 66.5% of the strength of the polystyrene

films (**e**), respectively, the HCC-PS film exhibited a significant increase in elongation (**f**) and toughness (**g**). Specifically, the elongation (**f**) and toughness (**g**) of the HCC-PS film were found to be 2.3 and 2.2 times higher than that of the polystyrene film, respectively, highlighting the unique and potentially valuable properties of the upcycled bioplastics. Data are presented as mean values ±standard deviations ($n = 12$ independent samples for PS, 6 independent samples for WT-PS, and 8 independent samples for HCC-PS). ** indicates $p < 0.01$. ns indicates no significance. Two-tailed Student's t-test was used. Source data are provided as a Source Data file.

A techno-economic analysis (TEA) was also conducted to evaluate the economic outlook of the RUMBA-based wastewater treatment and bioplastic production, assuming an annual production of 500 metric tons of bioplastic[45]. Economic assumptions, equipment costs, labor costs, and capital costs are detailed in Supplementary Tables 3–7. The analysis compares the minimum selling price (MSP) of the bioplastic under two algae cultivation systems: an open pond and a photobioreactor, integrating results from the LCA case (Supplementary Fig. 13 with byproduct displacement). The analysis indicates an MSP of $3.58/kg for the open pond system (Supplementary Fig. 13). This price falls within the lower end of the reported range of $3.52 to $146/kg for bioplastics[43,45]. Such an MSP is competitive in the bioplastics market as common bioplastics like PHA (Polyhydroxyalkanoates), which has an MSP around $4 to $6 per kg[46]. The MSP of bioplastics from the open pond system is most sensitive to bioplastic yield and residence time, with material cost being the largest component in the MSP (Supplementary Figs. 13 and 14). When using photobioreactor cultivation, the MSP of bioplastic increases to $30.68/kg, which is not competitive with the current market[47], primarily due to the higher capital investment required for photobioreactor systems (Supplementary Fig. 14). It is worth noting that current PBR systems are not scaled for environmental applications. Open pond system is actually more relevant to the algal bioproduction and wastewater treatment process. A comparison with existing literature suggests this approach has the potential to lower the MSP relative to previous estimates[48]. Furthermore, the current estimate did not account for carbon credits and the environmental benefits of removing microplastics, which could further reduce the MSP of upcycled plastics. Overall, the results indicated that the

synergy of microplastics removal, upcycling, and algal bioproduction could lead to economically viable microplastics remediation strategies.

## Discussion

RUMBA represents a conceptual design where synthetic biology-enabled microplastic bioremediation and upcycling can be integrated with various valorization routes. Microplastic removal was achieved through mechanism-guided rational engineering of cyanobacteria to enhance cell hydrophobicity, facilitating cell-microplastic interactions. Unlike EPS-based microplastic bioremediation from previous studies[24,25], which require EPS development after exposure to microplastics, RUMBA removes microplastics within minutes of HCC exposure to microplastics. The removal efficiency reaches 91.4% in 1 h and can be further enhanced with extended time. Moreover, RUMBA has been demonstrated to be effective in removing various types and sizes of microplastics, including PS, PE, and PET. The environmental sample analysis also indicated that RUMBA has the potential application to a broader spectrum of microplastics not tested in this study.

The RUMBA system is built upon UTEX 2973, a rapidly growing cyanobacterium strain with substantial potential for industrial applications, including biomass, sugar, and chemical production[49–51]. Furthermore, this strain has been proposed for use in wastewater treatment and nutrient removal[29]. Owing to these characteristics and the natural capacity of cyanobacteria for $CO_2$ utilization, RUMBA offers several unique advantages in sustainability compared to other remediation systems. First, the growth of cyanobacteria consumes $CO_2$, which could offset emissions during the microplastic remediation

process, making the process less emission-intensive or even carbon-negative. Second, RUMBA can be integrated with wastewater treatment to remove nutrients and microplastics simultaneously, further extending the environmental benefits of the process. Third, the integrated process not only removes microplastics but also yields cyanobacterial biomass, which can be further processed to generate various value-added products, enhancing the overall economics of the RUMBA. The microplastic removal, nutrient removal, and biomass production can be integrated and further optimized by tuning the treatment conditions to prioritize the process priority. For example, when integrating with wastewater treatment, it was observed that extended treatment time resulted in higher nutrient removal but reduced microplastic removal efficiency (Table 2). Thus, continuing to optimize the treatment time and closely monitoring nutrient compositions could potentially yield an optimal balance between nutrient recycling and microplastic removal. Moreover, it was also observed that supplying additional nutrients to wastewater promoted nutrient consumption, microplastic removal, and biomass production, likely due to increased cell hydrophobicity and cell density under nutrient-rich conditions. Further tuning the extra nutrient composition might yield more optimized nutrient and microplastic removal along with economics and sustainability.

In particular, a valorization pathway is demonstrated to upcycle microplastics using cyanobacterial biomass and microplastics yielded from RUMBA through bioplastic composite production. The bioplastic composite films were produced using sediments containing PS microplastics and cyanobacterial biopolymers. The composite showed enhanced elongation and toughness properties compared to pure PS films (Fig. 5), indicating their potential as sustainable alternatives to pure petroleum-based plastics. This strategy presents a synergistic concept by adding value to microplastic remediation and potentially enabling a circular economy.

While this study establishes the significant potential of RUMBA, several avenues for improvement are critical for its large-scale implementations. Future work should focus on optimizing the upcycling process for complex environmental scenarios, which often feature low concentrations and diverse mixtures of microplastics and other impurities. For instance, systematic studies are required to evaluate how conventional pollutants, such as heavy metals or organic contaminants, might impact HCC viability and the efficiency of the hydrophobic interaction mechanism, in addition to managing factors like COD in broader wastewater applications[52–55]. Furthermore, a deeper investigation into the roles of specific biopolymers from the cyanobacteria is key to systematically optimizing the composite's mechanical properties, potentially by fine-tuning process parameters like cell-to-plastic ratios or residence times. On the biological front, further synthetic biology efforts could enhance microplastic-HCC interaction, boost removal capacity, and improve overall bioproductivity. Crucially, while the system demonstrated robust performance over the 19-day trial, long-term industrial application necessitates a focus on stability. The effort would involve both process development to maintain optimal system conditions for consistent performance and further strain engineering to ensure genomic stability. The combined efforts can reduce the risk of critical mutations that could compromise the hydrophobic phenotype over extended operational periods. By addressing these challenges, RUMBA can evolve from a promising concept into a versatile and robust platform, effectively integrating microplastic removal with broad environmental remediation and downstream valorization.

## Methods

### Strains and growth condition

The wild type *Synechococcus elongatus* UTEX 2973[56] was generously gifted by Dr. Himadri B. Pakrasi from Washington University in St.

Louis, where extensive genomics information is available to reveal the rapid growth mechanism[57]. The limonene-producing HCC strain was developed as previously described, where a synthetic promoter has driven the ultra-high level limonene production, leading to a hydrophobic cell surface and auto-sedimentation[27,28]. Both strains were maintained in 250 mL Erlenmeyer flasks containing 50 ml of BG11 medium (Sigma, C3061) supplemented with 10 mM TES buffer (pH 8.2) and grown under a photon flux density of 50 μmol photons $m^{-2}$ $s^{-1}$ at 30 °C. For large-scale cultivation, Strains were cultivated in 1-L Roux bottle containing 500 ml of BG11 medium. These cultures were maintained at 37 °C and sparged with 5% (v/v) $CO_2$ in air at a flow rate of approximately 0.8 L/min. Fed-batch cultivation was performed every 24 h with 1× BG11, unless otherwise specified, starting from the second day of cultivation. This process continued until the $OD_{730}$ reached 15, where the late log phase is often also used for algal harvest[27]. This concentration thus could be used to synergize algal bioproduction, microplastic removal, and downstream upcycling. Concentration also ensures a high cell density while maintaining strong aggregation capacity. The temperature was maintained at 37 °C throughout the cultivation, and the light regime was set to one-side 357 μmol $m^{-2}$ $s^{-1}$, one-side 574 μmol $m^{-2}$ $s^{-1}$, and double-side 574 μmol $m^{-2}$ $s^{-1}$ for 0 - 12 h, 12–36 h, and thereafter.

### Microplastic removal assay

The following microplastics were used in this study: polystyrene (PS) nanoparticles of 200 nm (Cat# 69057), 500 nm (Cat# 59769), and 800 nm (Cat# 65984) from Sigma-Aldrich; polyethylene terephthalate (PET) particles ( < 300 μm, Cat# 1000080955) from Goodfellow; and polyethylene (PE) particles (32-38 μm, Cat# DNP-RPE03) from CD Bioparticles. The PS particles were supplied as aqueous suspensions. The PET and PE particles were supplied as powders, which were weighed on an analytical balance (Sartorius) before being added directly to treatment samples. The selection of these specific sizes was based on several considerations. The chosen sizes reflect the general distribution observed in wastewater treatment plants and acknowledge the inherent difficulty in removing smaller, nano-sized particles[11,58]. To evaluate the efficacy of hydrophobicity-mediated aggregation and sedimentation across different particle sizes, PS microplastics (200, 500, and 800 nm) were spiked into water samples. The final concentration of 0.02% (or as otherwise specified) is consistent with previous studies[9,59] and allowed for a robust test of the removal mechanism with particle sizes relevant to environmental samples.

Cyanobacterial cultures were harvested at an $OD_{730}$ of approximately 15. For removal assays, cells were mixed with PS microplastics to the designated final concentrations. For 200 nm PS, experiments were conducted in a total volume of 20 mL. For 500 nm and 800 nm PS, a total volume of 6 mL was used. The mixtures were gently inverted about 10 times to ensure homogeneity and then allowed to settle undisturbed for 1 h at room temperature. To evaluate the microplastic removal in wastewater and surface water environments, harvested cyanobacterial cells were gently concentrated by low-speed centrifugation (1000 × g for 10 min). The resulting pellet was then gently resuspended in the wastewater or surface water samples spiked with commercial or environmentally sourced microplastics.

The interaction between cyanobacteria and various microplastics was qualitatively assessed using fluorescence and brightfield microscopy. Wild-type or HCC cells from cultures at an $OD_{730}$ of ~15 were mixed with either PET, PE, PS (10 μm, Sigma 61946), or microplastics isolated from environmental water samples. The mixtures were mixed gently by inversion to facilitate interaction and incubated for about 30 min to allow aggregation. Following incubation, a 10 μL aliquot was carefully withdrawn from the bottom of the tube using a wide-bore

(cut-off) pipetted onto a glass slide and covered with a coverslip. For commercial microplastics (PET, PE, and PS), samples were imaged using a Leica DM6B fluorescence microscope. Cyanobacterial cells were visualized by their characteristic red chlorophyll autofluorescence using a Cy5 filter set. The intrinsic blue-cyan autofluorescence of PET and PE microplastics was observed using a DAPI filter set. For environmental samples, interactions were observed using a Boreal bright-field microscope (Boreal Optical, China) equipped with a Moticam X5 Plus camera (Motic, Richmond, BC, Canada).

The long-term stability of microplastic removal was evaluated in a semi-continuous fed-batch system. The initial culture was established in a 1-L Roux bottle containing 500 mL of 2× BG11 medium, spiked with 0.05 g/L of 800 nm PS microplastics, and fed daily with 1× BG11 medium. Cultivation conditions were identical to those described the "Strain and Growth condition" section. After an initial 4-day cultivation period, 50 mL aliquots ($n = 3$) were aseptically collected for analysis. The remaining culture was then used to inoculate the subsequent batch. To begin the next cycle, the $OD_{730}$ of the remaining culture was measured to calculate the required inoculum volume. This volume was transferred to a new Roux bottle containing fresh 2× BG11 medium to achieve a starting $OD_{730}$ of approximately 2.0. The new culture was re-spiked with 0.05 g/L of 800 nm PS microplastics. This reset and cultivation process was repeated, with subsequent sampling occurring every three days up to a total of 19 days. A similar protocol was applied using wastewater influent or effluent as the growth medium to assess the integration of microplastic removal with wastewater treatment.

## Suspension analysis for PS microplastic removal by cyanobacteria

After settling, aliquots from the top of the suspension were analyzed. Turbidity and chlorophyll fluorescence were measured in 200 μL samples in a 96-well plate using a SpectraMax iD5 microplate reader (Molecular Devices). Turbidity was measured by absorbance at 850 nm (a standard wavelength for turbidity sensors due to its minimized color interference[60]), and chlorophyll fluorescence was measured with an excitation wavelength of 595 nm and an emission wavelength of 690 nm, which was optimized by a SpectraMax iD5 microplate reader. All samples were diluted with BG11 medium as needed to maintain absorbance readings within the linear range of the instrument (0.3–0.7).

Differential centrifugation was applied to separate cyanobacterial cells and PS microplastics and to estimate the remaining PS microplastics in the suspension. Centrifugation speeds were set to 800 × g for samples containing 200 nm microplastics and 300 × g for 500 and 800 nm microplastics, and the centrifugation time was set to 3 min. The centrifugation procedures were effective in achieving good separation (Supplementary Fig. 3), which validated the reliability of the protocol. The relative turbidity (850 nm) of PS microplastic was calculated by normalizing the turbidity of HCC samples to that of WT control samples to account for any cell pelleting during centrifugation.

## Thermogravimetric analysis

TGA was applied to analyze the composition of mixture containing PS microplastics and cyanobacterial biomass. Approximately 5 mg of each mixture sample was loaded into a Thermogravimetric analyzer (Pyris 1 TGA, PerkinElmer) with a heating program that increased the temperature from 50 °C to 700 °C at a rate of 10 °C/min and then held at 700 °C for 10 min. Pure PS and cyanobacterial biomass were also analyzed as controls for calculation purposes. The weight at 150 °C in the output data for each sample was normalized to 100% to eliminate the impact of moisture. The percentage of PS and cyanobacterial biomass in the samples was calculated using Eqs. (1) and (2), respectively, based on the properties that PS loses 97.1% of its weight at 440 °C, while biomass retains 35.4% of its weight at the same

temperature (Supplementary Fig. 15).

$$\% Polystyrene_{in\ sample} = \frac{Pure\ Biomass_{T440} - Sample_{T440}}{Pure Biomass_{T440} - Pure\ Polystyrene_{T440}} \times 100 \quad (1)$$

$$\% Cyano_{in\ sample} = 100 - \% Polystyrene_{in\ sample} \quad (2)$$

$$Removal\ Efficiency =$$
$$100 - \frac{\% polystyrene_{in\ suspension} \times Suspension\ Dry\ Weight}{Total\ Microplastic\ Dry\ Weight} \times 100 \quad (3)$$

The $Sample_{T440}$, Pure Biomass$_{T440}$, and Pure Ploystyrene$_{T440}$ represent the percentage of the weight that is retained at 440 °C in mixture samples, pure cyanobacterial biomass samples, and pure PS microplastic samples, respectively. The % Polystyrene $_{in\ sample}$ and % Cyano $_{in\ sample}$ denote the weight percentage of PS and cyanobacterial biomass in the mixture samples, respectively. The % Polystyrene $_{in\ suspension}$ refers to the dry weight percentage of PS microplastics present in the suspension fraction after sedimentation. The Suspension Dry Weight was measured by weighing the dry mass after lyophilizing for 36 h.

## Electron microscopy imaging analysis

Transmission electron microscopy (TEM) was employed to observe interactions between cyanobacterial cells and PS microplastics. To prepare the samples, cyanobacterial cells were mixed with PS microplastics and incubated for 30 minutes to ensure thorough interaction before gently resuspending the sediments and taking samples. Droplets containing 2 μl samples were added onto copper grids and left for 5 min to allow cell and particle settling. The remaining liquids were removed with filter paper before negative staining with 1% uranyl acetate for 20 s. The grids were then air-dried for a few hours and observed under JEOL 1200. The SEM samples were lyophilized for 48 h before coating and the analysis was conducted using a Tescan FERA-3 Model GMH Focused Ion Beam Microscope at an accelerating voltage of 5 kV.

## Chemical imaging analysis

SRS microscopy described previously for plant biomass imaging was used to perform the chemical imaging[61]. Specifically, a HighQ pico-TRAIN (Spectra-Physics) laser was used to generate 1064 nm (up to 15 W) and 532 nm (up to 9 W) output. Both are pulse trains at 7 ps. The 1064 nm output was used as the SRS Stokes beam. The 532 nm beam was used to pump an APE optic parametric oscillator (Levante Emerald, APE GmbH, Germany) to produce a tunable wavelength 6 ps pulse train to be used as the SRS pump beam. The 1064 nm Stokes beam was modulated by an acoustic optical modulator (3080-122, Crystal Technology) at 10 MHz frequency, achieving >80% intensity modulation depth. Both the pump and Stokes pulse trains were combined (1064dcrb, Chroma) and routed to a modified scanner (BX62WI/FV300, Olympus) attached to an Olympus IX81 microscope. The pump beam intensity after the sample was collected by a high numeric aperture lens, filtered and detected by a photodiode. A lock-in amplifier was used to detect the stimulated Raman loss signal. The Raman frequency of limonene C=C bond at 1670 cm$^{-1}$[62,63] was chosen to image limonene, which corresponds to a pump wavelength at 903 nm. The Raman frequency PS $CH_2$ vibration at 2900 cm$^{-1}$[64,65] was used to image PS, which was corresponding to a pump wavelength at 813 nm.

## Zeta potential measurement

The zeta potentials of cyanobacterial cells and PS microplastics were measured using a Zetasizer Advance Pro (Malvern Panalytical). For the analysis, aliquots of both cell and microplastic suspensions were diluted with ddH$_2$O to a suitable concentration and maintained at

room temperature. To minimize potential changes in cell surface hydrophobicity following harvesting, cyanobacterial samples were analyzed within 3 h of removal from the growth culture.

## Environmental microplastics sampling process

To collect microplastics from environmental water samples, approximately 200 liters of water from the sampling sites were pumped into a series of layered sieves with pore sizes of 25 μm, 125 μm, and 500 μm. In the event that any sieves became blocked during the collection process, the sieves were promptly removed, and the collection procedure was continued until the target volume of 200 liters of water was reached. The collected particles were then transferred into a beaker, where saturated NaI was added for density separation. After complete dissolution of NaI, the sample was transferred into a separatory funnel and allowed to settle overnight, after which heavy particles in the sediment were removed. The remaining particles in suspension were collected using a vacuum filter device with a pore size of 0.45 μm. After washing with ddH$_2$O to remove salts, Nile Red stain was then performed on the filter to aid in the identification of Microplastics under a microscope. After staining, the particles were washed and resuspended with ddH$_2$O and transferred into a glass petri dish.

## Nutrient assay

Commercial kits from Hach Company were used to measure nitrate (TNT 835), phosphate (TNT 843), and ammonium (TNT 830) concentrations. All samples were properly diluted to the recommended measurement ranges and processed following the manufacturer's instructions. A Hach DR3900 benchtop visible spectrophotometer was used for quantification. Additionally, a nitrate ion-selective electrode was employed to monitor nitrate concentration over time during the integrative experiment, which combined wastewater nutrient removal and microplastic removal.

## Bioplastic film preparation

The sediments acquired from mixtures containing 35 mL of cyanobacteria and 0.3% w/v PS microplastics settled for 1 h were used for bioplastic composite production. The sediments were further concentrated by centrifugation before being lyophilized for 48 h until completely dry. The dry sediment was then dispersed in chloroform using sonication (10 s on/50 s off for 6 cycles) and centrifuged at 5700 × g for 10 min to separate major pellets. The remaining supernatant (containing some undissolved particles) was filtered through 0.2 μm PTFE filters (requires multiple filters to complete the filtration due to filter blockages) and then cast onto aluminum plates. To make the control pure PS film, PS foam was dissolved in chloroform and cast using the same procedure.

## Mechanical property tests for the films

The films were cut into strip-like shapes, with a dimension of about 1.5 mm in width and 3 cm in length. The precise width (*W*) and thickness (*T*) were measured using a Vernier caliper and a Spiral micrometer, respectively. The strips were glued onto a paper board sample holders and then fixed on the grippers. The original length (*L*) of strips was measured using a vernier caliper. A 100 Series Modular Universal Test Machines equipped with a 2 N load cell (TestResources, MN, US) was used to measure mechanical properties. The applied force (*F*) and corresponding displacement (*d*) were monitored synchronously during the measurement. The cross area (*A*) was calculated using the equation of $A = W \times T$. Stress−strain curves can be plotted after getting stress (*σ*) and strain (*ε*) using the equations of $\sigma = F/A$ and $\varepsilon = d/L$, respectively. The ultimate tensile strength was the maximum stress before fracture and the modulus of elasticity (MOE) was obtained from the slope of the elastic deformation region in the stress−strain curve. Elongation (%) was calculated using the equation of $d'/L \times 100$, where *d'* is the displacement at the fracture. The toughness was the integral area

under the stress-strain curve calculated using the Origin software (OriginPro 2021, US).

## Techno-Economic and Life Cycle Analysis (TEA/LCA)

TEA and LCA were performed to evaluate the economic feasibility and environmental impact of the RUMBA process. The detailed methodology, system boundary, and assumptions for these analyses are provided in Supplementary Information (Supplementary Figs. 11, 14, Supplementary Tables 2–7). The scenario and sensitivity analyses, which identify the key factors that impact the environmental and economic performance of RUMBA process, are shown in Supplementary Figs. 12, 13, Table 8, and Dataset 1).

## Statistical analysis

All experiments were performed in at least triplicates. No statistical methods were used to predetermine sample size. The experiments were not randomized, and the investigators were not blinded to allocation during experiments and outcome assessment. No data were excluded from the analyses. Data are presented as mean ± standard deviation. Statistical significance between groups was determined using a two-tailed Student's t-test. A *p*-value of <0.05 was considered statistically significant. Statistical differences among treatment groups for the integration systems (Table 2) were evaluated using a one-way Analysis of Variance (ANOVA), followed by a Tukey's HSD post-hoc test to identify significant differences between specific pairs of means ($p < 0.05$).

## Reporting summary

Further information on research design is available in the Nature Portfolio Reporting Summary linked to this article.

# Data availability

All data used to generate main text and Supplementary Figs. is available from the corresponding author upon reasonable request. The corresponding author will respond to all reasonable requests within 15 business days. All data needed to evaluate the conclusions in the paper are presented in the paper and the supplementary materials. Source data are provided with this paper.

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

## Acknowledgements

The authors would like to thank Dr. Stanislav Vitha from the Microscopy Image Center (MIC), Texas A&M University, for his assistance with microscopy imaging. Y.Z. acknowledges the support from the Laboratory Directed Research and Development (LDRD) Program at NREL. S.Y.D., J.S.Y., and C.F. acknowledge support from the US Department of Energy (DE-FE0032108). J.S.Y. and S.Y.D. acknowledge support from the US National Science Foundation (NSF ERC 2330245). S.Y.D. also acknowledge support from the US National Science Foundation (NSF MCB 2229160).

## Author contributions

B.L., Q.L., and S.Y.D. designed the experiments. B.L. and Q.L. carried out the microplastic removal experiments. C.H. did the mechanical testing for the bioplastics. Y.Z. performed the SRS chemical imaging analysis. W.L. carried out the zeta potential experiment. Y.C., M.L., and C.F. carried out TEA and LCA analysis. S.P. carried out the microplastic isolation experiments. B.L., Y.C., S.Y.D., and J.S.Y. discussed the results and co-wrote the manuscript. All authors contributed to the scientific discussions and comments on the manuscript.

## Competing interests

The authors declare no competing interests.
