## [Transparent Peer Review file · Nature Communications]

Remediation and Upcycling of Microplastics by Algae with Wastewater Nutrient Removal and Bioproduction Potential

Corresponding Author: Dr Susie Dai

Version 0:

Reviewer comments:

Reviewer #1

(Remarks to the Author)

This manuscript provides an interesting strategy to remove microplastics from water systems, including wastewater. However, it has a significant amount of generic information, benefiting from a more focused description of the content. In terms of research approach, the study fails in (undoubtedly) demonstrating the mechanisms involved in the removal process. The methods used are insufficient to demonstrate the role of hydrophobicity in the process. The same is valid for the role of limonene.

Looking into the process, it is unclear if the COD will be severely affected from including this step. Of additional concern is the microorganism selected. It is a cyanobacterium typically involved in algal blooms. Therefore, the translational relevance of the study fails, based on the lack of evidence on the potential of the cyanobacterium for wastewater treatment/polishing.

Reviewer #2

(Remarks to the Author)

The manuscript presents an innovative microplastic remediation system, RUMBA (Remediation and Upcycling of Microplastics by Algae), and demonstrates its effectiveness in microplastic removal, wastewater treatment, and microbial upcycling. The research exhibits high levels of innovation and application potential, offering a promising solution for microplastic pollution control and wastewater treatment processes. Overall, the manuscript presents promising research results, but there are certain limitations need to be improved. It is recommended that the authors enhance the discussion of the system's economic feasibility, scalability, and long-term stability in the manuscript, which could increase its scientific and practical value.

Specific comments:

1. The multifunctional remediation system proposed in the manuscript, which combines microplastic removal, wastewater treatment, and algae-based upcycling, is highly innovative. However, the introduction lacks a review of existing technologies. Please add a discussion that compares the proposed method with other similar technologies (e.g., traditional microplastic removal or biodegradation techniques), highlighting the differences, advantages, and disadvantages of the approach, and clarifying the strengths of this research.
2. Temperature and pH also effected the performance of cyanobacteria capturing microplastics, and corresponding environmental parameters need to be added in Table 1
3. The omics data of engineered cyanobacteria should be further clarified.
4. Figure 1g, is there significant difference obtained in the data of WT and HCC? Please add statistical analysis.
5. The microplastic removal efficiency and the system's upcycling performance are well-supported in this manuscript. However, there is a lack of detailed data on the system's long-term operational stability, particularly regarding the long-term impact of microplastic treatment. It would be beneficial to add data on the reliability, stability, and environmental impact of the system over extended periods of operation.
6. Please discuss the applicability and economic feasibility of the RUMBA technology for long-term operation in real wastewater treatment plants in the manuscript.
7. The manuscript could benefit from a more comprehensive explanation of how the microplastic removal efficiency and HCC upcycling performance can be optimized. I recommend further discussion of these factors in the interpretation of the results to provide more insight into potential improvements.
8. It is suggested to explore applications of the RUMBA system under different environmental and water quality conditions, and discuss whether the presence of conventional pollutants in the wastewater treatment system affects RUMBA's

performance.

9. Please provide an explanation of the meaning of L524 and ZM in Figure 2.

10. Please carefully check the manuscript for errors, such as the reference to Figure 2h in Line 158, which should be Figure 1h, and the reference to Figure 4 in Lines 213-215, which should be Figure 3. Please also carefully verify and correct the figure numbering in the Supplementary Information to ensure consistency with the main text.

11. Please add Life Cycle Assessment (LCA) and Techno-economic Analysis (TEA) in your manuscript. The environmental benefits and economic feasibility of RUMBA can be evaluated.

12. Was Supplementary Table 1 missed in the supplement?

Reviewer #3

(Remarks to the Author)

The manuscript titled "Remediation and Upcycling of Microplastics by Algae (RUMBA) Synergizing with Wastewater Nutrient Removal and Bioproduction" presents an innovative approach to addressing the critical environmental challenge of microplastic pollution. The authors attempt to integrate synthetic biology methodologies, wastewater treatment, and sustainable upcycling, tackling an issue of significant environmental relevance.

Although the manuscript explores an important and promising topic, it presents several shortcomings that undermine its robustness and scientific credibility. Significant revisions are required, particularly regarding statistical analysis of the data, contextualization within the existing literature, justification of methodological choices, and consistency in data presentation. The manuscript can only be considered for publication after addressing these issues.

General Comments:

1. The statistical treatment of the data must be performed and presented. Vague expressions such as "Substantial higher microplastic removal rate" or "centrifuged suspension were substantially lower in..." lack scientific meaning. The discussion of the results and conclusions must be revised based on statistical analysis. The absence of robust statistical analysis and more detailed data presentation undermines the credibility of the results.

2. The authors must justify the use of genetically modified cyanobacteria instead of naturally occurring species and discuss the implications of this choice. A detailed comparison with existing literature is necessary. A complete characterization of both strains (including the wild type) must also be provided.

3. The study addresses a highly relevant environmental issue with potential impact on sustainability and wastewater treatment fields. However, the lack of detailed justifications and a stronger contextualization within the existing literature limits its current scientific contribution.

4. The authors must justify the selected concentrations and sizes of the microplastics used, considering the characteristics of the water samples studied, and provide robust support for the experimental and analytical conditions employed.

5. Basing microplastic sedimentation on the hydrophobicity of limonene in HCC requires confirmation of the absence of EPS production and an analysis of the charges of the cells and microplastics. These data are crucial and must be provided.

6. The methodology shows significant gaps. The lack of justification for the experimental conditions and tested parameters undermines the study's robustness. Additionally, the described methods do not provide sufficient detail to ensure the reproducibility of the work.

7. The entire manuscript requires a thorough review, as it contains numerous errors, including issues with figure numbering, language inconsistencies, and methodological gaps.

Specific Comments:

Line 30 and others: Use impersonal language throughout the manuscript and supplementary materials.

Line 32: Consider replacing "new technology" with "system."

Line 78: Provide the zeta potential of the cells and microplastics.

Line 82: Improve the state of the art. Explain how limonene facilitates the self-aggregation of microplastics. Is limonene present on the surface of the cells? Does the cyanobacterium used not produce EPS? Are there natural species that produce limonene? What is the WT (Wild Type)? Provide a complete characterization of both strains, including the WT.

Line 117: Perform statistical analysis to support the conclusions.

Line 119-123: Since turbidity and fluorescence depend on the species, these analyses should be normalized using a control.

Line 126: Provide the density values for microplastics of different sizes.

Line 146-149: Explain the statistical significance of the observed differences.

Line 149: Present TGA results with replicates and statistical analysis.

Line 158: Review figure numbering. Justify the use of 5 mg of microplastics and indicate whether these concentrations reflect environmental conditions.

Line 200: Microplastics often adhere to glass walls or aggregate among themselves and sediment. To avoid this, Tween 20 is added for dispersion. Does this suggest that hydrophobicity is the driving force?

Line 222-223: Confirm the GPS coordinates.

Line 252: Review figure numbering.

Line 262: Explain the term "environmentally relevant conditions."

Line 279: Critically justify the tested concentrations and whether they are environmentally relevant.

Line 324: Evaluate the sustainability of the process used.

Line 406: Justify why an OD730 of 15 was used. Does this value represent the stationary phase of the species?

Line 418-419: Justify the choice of wavelengths used.

Line 426: Explain how the authors account for the turbidity effect of cyanobacteria.

Line 523: Add a section on statistical data analysis.

Figures and Tables:

Fig. 2: Review the nomenclature given to the samples. Ensure consistency in the nomenclature throughout the manuscript and supplementary materials.

Line 689: Replace "polystyrene microplastics" with its acronym. Review all similar cases in the manuscript.

Fig. 5: Review the nomenclature given. Ensure consistency throughout the manuscript.

Table 1: Add information to the main rows for clarity. Complete the missing references.

Table 2: To enhance clarity, add "Nitrate removal (%), microplastic removal (%), and biomass production (%)" to the table. Provide statistical analysis. Explain what the "integrated system" refers to.

Supplementary Materials:

Supplementary Figures: Revise the formatting to match the journal's standards and ensure correct citation in the text.

Figure 1: Provide statistical analysis between series and time points.

Line 8: Review to avoid repeating units in consecutive numbers.

Line 10: Revise the description of the sedimentation height.

Line 15-20: Revise and clarify the confusing text.

Line 27-29: Based on the images provided, the results appear to suggest the opposite. Revise.

Figure 4: Revise the figure legend. Expressions like "Substantial higher microplastic removal rate" lack scientific meaning and should be replaced with statistically supported analyses. The statement "b. sedimentation was observed in the HCC samples but not in the WT samples" is not evident from the provided images. It is recommended to review the images, comparing controls and PS samples. Ensure that sample nomenclature is consistent throughout the figure and text.

Figure 6: The figure legend needs to be completed, providing the exact meaning of symbols such as **, +, and -, as well as the corresponding significance levels.

Figure 8: Specify the number of cultivation days to which the results refer. Additionally, statistical analysis between series and concentrations must be included to ensure the observed differences are adequately justified. In the phrase "...resulting in less contact with the cyanobacterial cells," consider whether reduced shading caused by the microplastics is the main factor. A detailed discussion of this possibility is recommended.

Figure 10: Provide statistical analysis and review the sample nomenclature.

Version 1:

Reviewer comments:

Reviewer #1

(Remarks to the Author)

My comments have been addressed by the authors.

Reviewer #2

(Remarks to the Author)

The article has addressed the issues I was concerned about. and this version of the manuscript could be acceptable.

Reviewer #3

(Remarks to the Author)

I have completed my evaluation of the manuscript titled Remediation and Upcycling of Microplastics by Algae (RUMBA): Synergizing with Wastewater Nutrient Removal and Bioproduction (NCOMMS-25-00425).

The authors have made substantial improvements to the text, figures, and methodological details, and have adequately addressed the majority of the critical issues raised in the previous review. Specifically, the following aspects were satisfactorily revised:

- inclusion of statistical analyses to support the results;
- clarification and justification of methodological choices;
- improved contextualization within the existing literature;
- correction of figure labeling and sample nomenclature;
- addition of data on zeta potential, EPS production, and hydrophobic interactions.

These revisions significantly enhance the scientific rigor, clarity, and overall quality of the manuscript. I consider the revised version suitable for publication in Nature Communications. Congratulations to the authors.

Reviewer #4

(Remarks to the Author)

The environmental impacts presented in Supplementary Table 7 are evaluated using the CML v4.0 (2016) methodology, and each impact category is reported with its corresponding functional unit. The inclusion of avoided burden credits (e.g., for electricity and plastic film displacement) is a reasonable approach for system expansion. However, the following clarifications should be provided:

(a) Details (life cycle emission factor for the displaced conventional plastic in kg CO₂-eq/kg conventional plastic) on what is displaced (e.g., grid mix, plastic).

(b) The value of how much bioplastic displaces conventional plastic on a functional equivalence basis.

(Table S2 and Figure S11) Supplementary Table 2 provides a useful summary of the primary process inputs and outputs per liter of wastewater.

The system boundary includes "algae storage" but does not quantify or discuss potential emissions from this stage (e.g., biomass degradation, VOC, or CH₄/N₂O emissions during wet storage). The authors are encouraged to either clarify that storage emissions are assumed negligible or to justify this simplification with supporting references or estimates.

There are also concerns regarding the parameter assumptions used in the techno-economic analysis, specifically:

(a) Operating labor costs are not explicitly reported in Supplementary Table 4. While the fixed capital investment typically includes installation labor, the lack of clarity regarding operating labor cost estimation may lead to an underestimation of annual operating expenses.

(b) The authors should clarify how indirect equipment costs (e.g., installation, instrumentation, piping) were estimated. Is it based on empirical parametric estimation/database/ tools or software to perform the TEA modeling? A recent article (DOI: 10.1016/j.cep.2024.110001) highlights how neglecting installation and labor-related indirect costs can lead to optimistic cost projections. I suggest the authors provide more details to ensure reproducibility of their economic assessment.

Version 2:

Reviewer comments:

Reviewer #4

(Remarks to the Author)

The authors have addressed the reviewer's comments and improved the quality of TEA and LCA in their manuscript.

Response to Referee's Letter

We would also like to thank all three reviewers for recognizing our work's potential and pointing out directions for improvement. We thus performed more experiments, conducted more analyses, and carried out a major revision of the manuscript based on all comments. The revisions are shown in the blue font in the edited manuscript. Our point-by-point responses are included below. It should be noted that this letter cited 20 references. However, in any quoted text, we kept the reference numbers as in the manuscript and did not include these references in the letter. For the responses not quoting the manuscript text, we have included the references, with the author's name and year, to distinguish from the references in the manuscript text.

Reviewer Comments

Reviewer 1

This manuscript provides an interesting strategy to remove microplastics from water systems, including wastewater. However, it has a significant amount of generic information, benefiting from a more focused description of the content. In terms of research approach, the study fails in (undoubtedly) demonstrating the mechanisms involved in the removal process. The methods used are insufficient to demonstrate the role of hydrophobicity in the process. The same is valid for the role of limonene. Looking into the process, it is unclear if the COD will be severely affected from including this step. Of additional concern is the microorganism selected. It is a cyanobacterium typically involved in algal blooms. Therefore, the translational relevance of the study fails, based on the lack of evidence on the potential of the cyanobacterium for wastewater treatment/polishing.

Response: Thank you for your insightful comments and for recognizing that the manuscript presented an interesting strategy to remove microplastics. We appreciate the opportunity to clarify all the other aspects of our research to address the concerns as follows:

1. Regarding “generic information” and “more focused description of the content”:

We appreciate your feedback regarding the manuscript's focus, which helps improve the readability. Regarding the generic information, we have improved our manuscript in multiple ways. First, for the **Introduction** section, we have provided 1) specific comparison of and relevant references regarding to state-of-the-art methodologies to remove microplastics, 2) specific comparison of and relevant references in bio-based methods to

remove microplastics and their mechanisms and limitations, 3) detailed discussion regarding how our unique microplastics valorization strategy is different from the state-of-the-art, supported by the relevant references based on the synthesis of multiple reviewers' comment. Second, in the **Results** and **Discussion** sections, we comprehensively studied and discussed the interaction between different types of microplastics and the engineered cyanobacteria cells, compared to the state-of-the-art. Furthermore, we characterized the cyanobacteria's interaction with the real microplastics in the wastewater samples, in the context of real environmental applications. To further improve the specificity and mechanistic insights, we carried out additional GC/MS analysis, BATH assay, Zeta potential, and turbidity analysis to nail down the mechanism relevant to hydrophobicity and limonene production. Third, we have carefully revised the entire manuscript by streamlining introductory and background sections, ensuring all presented information is focused and essential, not generic.

2. Regarding the research approach and reviewer 1's concerns over insufficiency to show the mechanism (hydrophobicity and limonene) in the removal process:

We thank you for your suggestions to provide more evidence to show the removal mechanism, particularly the roles of hydrophobicity and limonene. We have taken your feedback seriously, performed several experiments, cited more relevant literature, and provided thorough evidence to support our hypothesis that hydrophobicity is the primary driver for microplastics adsorption and removal, and limonene is driving the hydrophobicity. The additional experiments include direct hydrophobicity measurement (BATH assay), surface charge analysis (zeta potential), and comparative extracellular polymeric substances (EPS) analysis. The new experiments have demonstrated the significantly increased surface hydrophobicity of the engineered strain compared to that of the wild-type strain, the resulting surface charge change, and the microscopic evidence of the cell/microplastics interactions. This further enhanced our original presentation of TEM and Stimulated Raman Microscopy (SRS Microscopy), which provided the chemical characteristics of the cell surface. These additional characterizations provided a sound foundation for the mechanism to verify the hypothesis that limonene enhanced hydrophobicity, which enhanced the interaction and sedimentation with microplastics.

At the same time, limonene-induced hydrophobicity to drive cell self-aggregation has been well established in a peer-reviewed publication in Nature Communications (Long et al., 2022). In our original manuscript, we were trying to avoid redundant information and focused on the innovation on microplastics remediation. However, we agreed with the reviewer's perspective that this study is an independent study and the mechanisms should be illustrated in the context of microplastics removal. Thus, we have carried out comprehensive

key experiments and analyses to verify the hypothesis that limonene-induced hydrophobicity is driving the microplastics removal. These additional characterizations provided a sound foundation for the mechanism to verify the hypothesis that limonene enhanced hydrophobicity, which enhanced the interaction and sedimentation of microplastics. The complete set of data to elucidate the removal mechanism is summarized below:

- **The Verification of Enhanced Hydrophobicity in Engineered Strain:** To further verify the hypothesis, we have carried out **BATH (Bacterial Adherence to Hydrocarbons) assay**, which quantitatively demonstrates significantly increased cell surface hydrophobicity of the engineered HCC strain compared to the Wild Type (WT) (Figure 1g). This directly supports the foundational hypothesis that enhanced hydrophobicity drives the interaction. The BATH assay has been commonly used to characterize hydrophobicity across literature (Rosenberg et al., 1980).

Figure 1g. BATH assay results

- **Surface Charge Characterization:** We have further incorporated **zeta potential measurements**. These results (Figure 1h) show a notable difference in surface charge between HCC and WT cells, with HCC exhibiting a less negative zeta potential. This reduction in negative charge, potentially attributed to increased surface hydrophobicity, would decrease electrostatic repulsion and facilitate closer interaction with hydrophobic microplastics. The zeta potential changes of the engineered strain provided another level of evidence that hydrophobicity might be contributing to the interaction with microplastics.

Figure 1h. Zeta potential results

- Hydrophobic Interaction Disruption:** As presented in the original manuscript, the experiment using Tween 20, a surfactant, demonstrated that disrupting hydrophobic interactions significantly impairs the microplastic removal efficiency of HCC. This provides functional evidence for the critical role of hydrophobicity in the aggregation and co-sedimentation process.
- TEM verification of microplastics-microbial interaction:** As presented earlier, our **Transmission Electron Microscopy (TEM)** images visually show a preferential and enhanced attachment of microplastics to HCC cells, as compared to the wild-type cells. Moreover, the interactions are particularly at cell intersections where limonene is expected to accumulate (Figure 2a). The results highlighted that the HCC cells preferentially interact with microplastics. Further light microscope images of environmental microplastics also showed that HCC cells aggregated with the microplastics' surfaces, while the wild-type cells did not. All this data indicated that the surface of HCC cells can selectively interact with microplastics, while wild-type cells cannot. This provides indirect evidence that surface hydrophobicity might be important. Together with the BATH assay (new data), zeta potential (new data), and hydrophobicity interaction disruption assay, the study clearly defines that surface hydrophobicity is critical in the interaction with microplastics and achieving sedimentation.
- Stimulated Raman Scattering (SRS) microscopy provides chemical evidence:** the cutting-edge SRS-microscopy can provide precise chemical composition of the living organism. The imaging data (Figure 2b, lower panel) show the co-localization of limonene accumulation regions with areas where microplastics interact with the HCC cell surfaces. This data, together with the previous data, demonstrated that limonene has mediated the hydrophobic interaction that achieved microplastics removal.

- **Role of EPS:** To further clarify the mechanism, we carried out additional experiments to compare exopolysaccharides, a major component of extracellular polymeric substances (EPS) production, between WT and HCC strains (black color, wild-type, purple color, engineered strain HCC). Our findings indicate that the HCC strain produces fewer exopolysaccharides than those produced by the WT strain. This observation strongly suggests that the enhanced microplastic interaction by HCC is not EPS-mediated but rather driven by the engineered cell surface hydrophobicity due to limonene(Long et al., 2022).

Supplementary Figure 1g. Exopolysaccharides production in WT and HCC

Collectively, the combined evidence including the direct hydrophobicity measurement (BATH assay), surface charge analysis (zeta potential), visual and chemical microscopic evidence (TEM, SRS), functional disruption of hydrophobic interactions (Tween 20), and comparative EPS analysis together provided a robust and compelling demonstration of the hydrophobicity-driven mechanism, with limonene playing a key role in enhancing this surface property. The hypothesis validation is highly consistent with the previously published peer-reviewed studies(Long et al., 2022).

As the revisions are comprehensive and presented in the new version in many places (by blue font), we hereby presented the summary and the new figures coming from the manuscript in this response letter as above. These new data and figures are included in the supplementary materials and the main text.

3. Regarding the concern about Chemical Oxygen Demand (COD) as an outcome of the process:

We thank the reviewer for pointing out the need to consider COD when the microplastics remediation approach is used in wastewater treatment. Indeed, the impact on COD has been considered in many algae-based wastewater treatments. For example, many established and emerging algae-based wastewater treatment systems effectively manage

COD, often through symbiotic relationships with bacteria that degrade organic matter, and by transforming wastewater nutrients into algal biomass, which is subsequently harvested (Abdelfattah et al., 2023; Hammond et al., 2025). The inherent auto-sedimentation capacity of the HCC algal strain, central to our RUMBA system, has the potential to enhance COD removal (Samiotis et al., 2022), as an additional benefit of our platform. This could occur through the assimilation of dissolved or particulate organic matter from the wastewater into cyanobacterial biomass, which is then efficiently removed from the water via auto-sedimentation. This principle is similar to concepts explored in settleable algal-bacterial culture systems, demonstrated with efficient COD removal capacity (Su et al., 2011). The specific engineering solutions for COD management need to be developed when scaling and applying RUMBA in various wastewater treatment systems.

Even though the benefits of COD management through bacterial/algal systems have been demonstrated, we are not sure if it will be a fair and relevant comparison of RUMBA and other algal/bacterial wastewater treatment on COD management, as RUMBA will remove algal biomass together with microplastics from wastewater and upcycle to bioplastics. The process will remove algal biomass from wastewater and certainly reduce the total COD, while algal and algal/bacterial wastewater treatment is a very different process. In fact, numerous studies have established that COD can be effectively managed when *Synechococcus elongatus* is used for wastewater treatment and nutrient removal (Samiotis et al., 2021; 2022). Apparently, a full-scale wastewater treatment using algal biotechnology would carefully manage COD/BOD, as the wastewater treatment system is a unique ecosystem where COD and algal growth would be interconnected. The negative impact on COD often happens when the algal biomass decays or lyses during unwanted algal blooms, where is not the case in our RUMBA system. The RUMBA concept aims to fully utilize the algal biomass toward bioproduction of bioplastics materials (i.e., supported by our new technoeconomic (TEA) analysis). The primary focus of our manuscript is to introduce and validate the novel RUMBA concept and method for microplastic removal and upcycling, along with its synergistic potential for wastewater treatment. Given the potential benefit and recovery of the algal biomass and the resulted potential COD reduction, we feel that COD management (positive impact) is beyond the scope of this microplastics removal and valorization study. We hereby address the reviewer comments by adding a brief discussion of COD management, acknowledging that COD management would be a consideration for full-scale implementation RUMBA in wastewater treatment. In line 502 to 507, it reads,

“Future work should focus on optimizing the upcycling process for complex environmental scenarios, which often feature low concentrations and diverse mixtures of microplastics and other impurities. For instance, systematic studies are required to evaluate how conventional pollutants, such as heavy metals or organic contaminants, might impact HCC viability and

the efficiency of the hydrophobic interaction mechanism, in addition to managing factors like COD in broader wastewater applications^{52, 53, 54, 55}.”

4. Regarding the choice of microorganism and its translational relevance:

We appreciate the reviewer pointing out that strain selection is important for environmental applications. As a matter of fact, as researchers using cyanobacteria for environmental and energy applications, we are fully aware of the potential of the selected *Synechococcus elongatus* UTEX 2973 strain and the risks of other cyanobacterial strains in algal blooms. We thus have expanded the discussion to address it. In fact, there are many publications about the potential of UTEX 2973 in industrial applications due to this rapid growth (Roh et al., 2021; Yu et al., 2015; Zhang et al., 2020).

- **Algal Bloom Concern:** It is important to clarify that while *Synechococcus* is a genus of cyanobacteria, the specific strain used, *Synechococcus elongatus* UTEX 2973, is not associated with the formation of harmful algal blooms (HABs) in the same way as some other cyanobacterial species. For cyanobacteria, algal blooms are more often associated with genera *Microcystis*, *Anabaena* (*Dolichospermum*), *Aphanizomenon*, *Cylindrospermopsis*, and *Lyngbya* (Zhang et al., 2023). The UTEX 2973 strain does NOT contain microcystin producing genes. In fact, a recent study from our colleagues have indicated that UTEX 2973 can be used as a strain to manage the algal bloom, as it out-competes the species that produces microcystin, and thus reduce the toxicity when algal bloom happens (Lee et al., 2024).
- **Wastewater Treatment Potential:** Our study demonstrates the nutrient removal capabilities of the engineered strain from actual wastewater samples (as shown in Supplementary Table 1 and Table 2). This aligns with existing research highlighting the potential of *S. elongatus* to be used in nutrient removal and wastewater treatment (Lee et al., 2024; Samiotis et al., 2022). The integration of microplastic removal with these established benefits of cyanobacteria in wastewater treatment is a key advantage of the RUMBA process.

Overall, we appreciate the reviewer’s comments and expanded the discussion in Line 472 of Page 477 as follows,

“The RUMBA system is built upon UTEX 2973, a rapidly growing cyanobacterium strain with substantial potential for industrial applications, including biomass, sugar, and chemical production^{49, 50, 51}. Furthermore, this strain has been proposed for use in wastewater treatment and nutrient removal²⁹. Owing to these characteristics and the natural capacity of cyanobacteria for CO₂ utilization, RUMBA offers several unique advantages in sustainability compared to other remediation systems.”

Overall, we appreciate Reviewer 1's critical comments, which helped us improve our manuscript. We have carried out additional experiments and a major revision of the manuscript to address these comments. These revisions have helped us to provide more direct evidence to validate our research hypothesis and build a stronger case that the RUMBA system offers a promising and innovative approach to microplastic pollution. We are grateful for the feedback, which is very instructive.

Reviewer 2

The manuscript presents an innovative microplastic remediation system, RUMBA (Remediation and Upcycling of Microplastics by Algae), and demonstrates its effectiveness in microplastic removal, wastewater treatment, and microbial upcycling. The research exhibits high levels of innovation and application potential, offering a promising solution for microplastic pollution control and wastewater treatment processes. Overall, the manuscript presents promising research results, but there are certain limitations need to be improved. It is recommended that the authors enhance the discussion of the system's economic feasibility, scalability, and long-term stability in the manuscript, which could increase its scientific and practical value.

Response: We appreciate that the reviewer recognizes that our research "exhibits high levels of innovation and application potential, offering a promising solution for microplastic pollution control and wastewater treatment processes." We agree that enhancing the discussion on economic feasibility, scalability, and long-term stability would increase the manuscript's scientific and practical value. We have addressed these aspects in greater detail through additional experimental data collection, technoeconomic and environmental impact analyses, which are presented in the revised manuscript and detailed in the following responses.

1. The multifunctional remediation system proposed in the manuscript, which combines microplastic removal, wastewater treatment, and algae-based upcycling, is highly innovative. However, the introduction lacks a review of existing technologies. Please add a discussion that compares the proposed method with other similar technologies (e.g., traditional microplastic removal or biodegradation techniques), highlighting the differences, advantages, and disadvantages of the approach, and clarifying the strengths of this research.

Response: Thank you for this valuable suggestion. We have expanded the Introduction section (specifically in the second paragraph) to include a more comprehensive review of

existing microplastic remediation technologies. The specific section of the introduction now reads with the new edits in blue font:

“The 300 million tons of non-degradable petrochemical plastics disposed per year have presented significant environmental challenges¹. Besides low recyclability, the decomposed microplastics and nanoplastics pose major threats to ecosystems, with negative impacts on a broad spectrum of living organisms, including microorganisms, plants, animals, and human beings^{2,3,4}. In particular, a recent remarkable study shows that patients with microplastics present carotid artery plaque have a higher risk of composite outcomes of myocardial infarction, stroke, or death⁵. Recent studies indicated that microplastics and nanoplastics can accumulate up to 0.5% of brain biomass and could be associated with a series of health issues from dementia to Alzheimer’s(Campen et al., 2024). Conventional microplastics remediation methods involve filtration and flocculation, which are not only costly, but also subject to clogging and water chemistry variations to impact the effectiveness. The current state-of-the-art microplastic remediation still primarily focuses on designing different systems to remove the microplastics from the aqueous phase^{7,8,9}. For example, one study developed core-shell superparamagnetic nanoparticles that attract and glue various microplastics into larger agglomerates, which can then be removed from water with an external magnetic field⁹. Another study employs an interfacial solar evaporation platform to simultaneously produce clean water and significantly improve microplastic removal from the source¹⁰. While removing microplastics is important, such a strategy leaves a waste stream that needs storage or further processing of the spent media¹¹. Even though the current wastewater treatment systems showed varied capacities in microplastic removal¹², the removal capacity decreases when the microplastics sizes get smaller¹². It is therefore critical to advance economic, sustainable, and effective microplastic remediation technologies that could integrate with wastewater treatment and take considerations of end-of-life plastics' fate.

To address the challenges of cost and secondary waste, an ideal solution would integrate microplastic removal with a valorization strategy¹³. This reframes microplastics from an environmental hazard into a potential carbon-rich feedstock for upcycling, a key step toward a circular economy¹⁴. However, a significant technological barrier remains: the high cost and inefficiency of current separation methods make it unfeasible to generate a microplastic stream for conventional recycling¹². A paradigm shift is needed, moving from simple removal to a synergistic system that can both capture microplastics and simultaneously generate value. Here, biological systems offer unique opportunities. If an organism could not only sequester microplastics but also produce a valuable co-product, such as biomass or bioplastics^{15,16,17}, the entire process might become

economically viable. This approach would create a direct downstream pathway for the captured plastics, embedding them within the biomass for co-processing and eliminating the risk of re-release into the environment^{18,19}. While algal biomass has been explored for bioplastics production^{15,16,17}, no current research has proposed synergizing microplastic capture with the simultaneous manufacturing of a valuable algal biomass, a strategy that could finally overcome the economic and logistical hurdles of microplastic remediation.

*Among different microplastic remediation technologies, bioremediation is more environmentally friendly because microorganisms are renewable, easy to grow, and the method is compatible with various downstream processes. Conventional microbial biodegradation methods will depend on the type of plastics to be remediated and have limitations when applied to the environmental remediation where a variety of different types of microplastics exist(Cholewinski et al., 2022). An alternative mechanism involves cell surface adhesion, where recent studies have found a strong positive correlation between a microbe's extracellular polymeric substances (EPS) production and its microplastic removal capacity^{21,22,23}. For example, *Gloeocapsa* sp., which produces high levels of EPS, demonstrated excellent microplastic removal capacity, while poor producers such as *Microcystis panniformis* and *Synechococcus elongatus* PCC 7942 showed minimal MP removal^{21,23}. Furthermore, modulating *Pseudomonas aeruginosa* EPS production has successfully achieved 'capture and release' of microplastics^{24,25}. Using microbial EPS to remove microplastics is largely dependent on the species and their capacities vary significantly^{22,23}. Moreover, the EPS-based microplastic removal is relatively slow and inefficient (Table 1)^{21,22,24,25}. For example, it takes *Synechococcus* sp. PCC 7002 over six hours to achieve a modest removal rate (around 80%) of microplastics by the high EPS-producing strain²¹. Furthermore, the utilization of microbial microplastics remediation will always have to take into consideration of toxicity, as the previously mentioned strains like *Microcystis panniformis* are also notorious for producing microcystin to endanger human, livestock and wildlife health²⁶.*

*Despite the limitations, microbial microplastics removal could achieve economic and sustainable process if the organism can produce useful bioproducts, remediate other contaminants, and achieve high efficiency of microplastic removal. In this regard, cyanobacteria have the potential to synergize the excess nutrient utilization. In a previous work, it is showed that engineering a hydrophobic cell surface can promote cyanobacterial self-aggregation, thus facilitating the low-cost recovery of the biomass at a high efficiency(Long et al., 2022). The engineering design was achieved through computational modeling-guided synthetic biology design of high limonene production, and the base strain *Synechococcus elongatus* UTEX 2973 (UTEX 2973) is known for high productivity and absence of microcystin producing genes^{27,28,29}. Such hydrophobic-*

mediated cell-to-cell interactions and self-aggregation inspired us to explore whether hydrophobic effects could also drive interactions between hydrophobic cells and microplastics, given that most plastics are highly hydrophobic. This approach could not only offer an efficient and sustainable method but also introduce a novel mechanism for microplastic remediation.”

2. Temperature and pH also effected the performance of cyanobacteria capturing microplastics, and corresponding environmental parameters need to be added in Table 1.

Response: We appreciate this point. We have reviewed the literature cited in Table 1. Where available, we have added information on the temperature used in those studies to Table 1. However, the pH values in these studies have not been well documented so we decided to not include them in the table. Our study's conditions are detailed in the Table and Methods sections. The new Table 1 is included in the following page:

Table 1 Summary of microorganism-based microplastic capture.

Species	Mechanism	Design	Microplastic type and size	Temperature	Efficiency	Ref.
M. panniformis	EPS	N.A.	PS (<106 μm) and PMMA (<250 μm)	20 \pm 1 $^{\circ}\text{C}$	N.A.	23
Scenedesmus sp.						
Tetraselmis sp.						
Gloeocapsa sp.						
Cyanotheca sp.	EPS	N.A.	PS (0.1 and 10 μm)	25 \pm 1 $^{\circ}\text{C}$	N.A.	22
Synechococcus PCC 7942	EPS	N.A.	PS (0.1 and 10 μm)	30 $^{\circ}\text{C}$	~18% in 6.5h [#]	21
Synechococcus PCC 7002					~ 82% in 6.5h [#]	
P. aeruginosa	EPS	Knockout of wspF & inducible expression of yhh	PET, PMMA, nylon, PVC (< 106 μm), Microplastics from seawater (106 - 300 μm)	25 $^{\circ}\text{C}$ or 30 $^{\circ}\text{C}$	> 90% in 24h [#]	24
P. aeruginosa	EPS	Expression of trypsin & laboratory evolution	PS, PET, PMMA (< 106 μm), Microplastics from seawater (106 - 300 μm)	37 $^{\circ}\text{C}$	N.A.	25
Synechococcus UTEX 2973	Hydrophobicity	Expression of limonene synthase	PS (<5 μm), PET (<300 μm), PE (35 μm), Microplastics from surface water and wastewater.	37 $^{\circ}\text{C}$ and room temperature	91.4% in 1h	This study

N.A., not applicable or not indicated in the original paper; EPS, extracellular polymeric substances; PS, polystyrene; PMMA, polymethyl methacrylate; PET, polyethylene terephthalate; PVC, polyvinyl chloride; PE, polyethylene.

3. The omics data of engineered cyanobacteria should be further clarified.

Response: Thank you for your comment. We have clarified the genetic basis of the engineered cyanobacteria (HCC strain) in the manuscript. The HCC strain was engineered for limonene production as previously described in our cited work (Long et al., 2022), which details the synthetic biology design. We hereby expanded the detailed description of the strains and cited the relevant references (including two additional ones) to cover the omics and synthetic biology information. L523-L527 now reads:

*“The wild type *Synechococcus elongatus* UTEX 2973⁵⁶ was generously gifted by Dr. Himadri B. Pakrasi from Washington University in St. Louis, where extensive genomics information is available to reveal the rapid growth mechanism⁵⁷. The limonene-producing HCC strain was developed as previously described, where a synthetic promoter has driven the ultra-high level limonene production, leading to a hydrophobic cell surface and auto-sedimentation^{27,28}.”*

4. Figure 1g, is there significant difference obtained in the data of WT and HCC? Please add statistical analysis.

Response: We appreciate the comment and agree with the reviewer that statistical analysis is important. We have performed the statistical analysis (i.e., t-test) on the data presented in Figure 1g (now Figure 1e in the revised manuscript). The t-test has shown a P-value < 0.01. The figure and its caption have been updated to include significance indicators (e.g., p-values or asterisks) to denote significant differences between WT and HCC samples. The updated figure is as shown below.

Figure 1e

5. The microplastic removal efficiency and the system's upcycling performance are well-supported in this manuscript. However, there is a lack of detailed data on the system's long-term operational stability, particularly regarding the long-term impact of microplastic treatment. It would be beneficial to add data on the reliability, stability, and environmental impact of the system over extended periods of operation.

Response: This is an excellent point, and we appreciate the reviewer highlighting the importance of long-term operational stability. To address this, we have now included new data from a 19-day continuous microplastic treatment experiment. During the 19-day experiment, we found the system’s performance was stable. The 19-day experiment was conducted in a semi-continuous cultivation mode, encompassing 6 batches, without the new addition of seed cultures after the initial setup. Throughout this extended period, each batch demonstrated stable sedimentation with no significant reduction in sedimentation performance, indicating the potential for long-term stable and effective microplastic removal by the HCC strain (Supplementary Figure 1f). The MP removal efficiency was estimated to be consistently between 73% and 88% across this 19-day timeframe. This result, along with the fed-batch cultivation and microplastic removal results over 9 days (Table 2), provides more substantial insight into the system's operational performance and reliability over an extended period. We have also expanded the Discussion section to address the need for future research focusing on even longer-term operational stability and a more in-depth assessment of the environmental impact under continuous, scaled-up operation.

The added discussion section reads: *“Crucially, while the system demonstrated robust performance over the 19-day trial, long-term industrial application necessitates a focus on stability. The effort would involve both process development to maintain optimal system conditions for consistent performance and further strain engineering to ensure genomic stability. The combined efforts can reduce the risk of critical mutations that could compromise the hydrophobic phenotype over extended operational periods.”* in L512-517.

The new supplementary Figure 1f is as follows to demonstrate long-term stability.

Furthermore, to assess the environmental impact, we have carried out Life Cycle Analysis (LCA). Our LCA (Supplementary Figure 12) shows RUMBA can achieve significant negative CO₂ emissions up to -6.63 kg CO₂ emissions per kg of bioplastics produced (considering byproduct displacement), while providing nutrient and microplastic removal benefits. This compares favorably with conventional bioplastics.

Supplementary Figure 12

We also added a paragraph in the main text to discuss the LCA for the RUMBA process (Line 412-436). The paragraph reads: *“The techno-economic Analysis (TEA) and life cycle analysis (LCA) were carried out to evaluate the feasibility and scenario for commercial application of RUMBA, along with its emission impacts (Supplementary Table 2-7 and Supplementary Figure 11-14). LCA was performed to evaluate the environmental impact of the RUMBA process integrating with algae cultivation, microplastic removal, wastewater treatment, and bioplastic production. The functional unit, system boundary, and inventory analysis are detailed in the Supplementary Figure 11 and Supplementary Table 2. If the system is powered by conventional energy, the total emission ranges from 19.38 to 20.67 kg CO₂ equivalent per 1 kg of bioplastics, with electricity consumption remaining to be the primary contributor to CO₂ emissions across the RUMBA-based wastewater treatment and bioplastic production life cycle (Scenarios 3 and 4 in Supplementary Figure 12). To promote a circular economy for algae, we also consider the use of renewable energy to power the system⁴¹, which is very feasible as the algal production can be readily coupled with solar energy as both need sunlight. Without considering residual biomass allocation, the production of 1 kg of bioplastics through upcycling results in a net utilization of 3.21 kg CO₂ emissions using renewable power sources. The actual industrial implementation will certainly carry out the byproducts displacement, and the scenario analysis highlights the potential for this process to achieve negative emissions (-4.50 kg CO₂ emissions/kg of bioplastic produced when considering residual biomass for electricity generation). This stands in contrast to other bioplastic production methods, which typically have positive CO₂ emissions ranging from 0.6 to 282.6 kg CO₂ emissions/kg of bioplastic produced^{42, 43}. In addition to impacts on*

greenhouse gas emissions, this process confers positive environmental impacts by utilizing nitrogen and phosphate from wastewater and removing microplastics⁴⁴. The results highlighted that the synergy of microplastics removal and upcycling, algal production, and wastewater nutrient usage could achieve a significant positive environmental impact, as shown in Supplementary Table 7.”

6. Please discuss the applicability and economic feasibility of the RUMBA technology for long-term operation in real wastewater treatment plants in the manuscript.

Response: We appreciate the comments and agree that discussing applicability and economic feasibility is crucial. We have performed a detailed Techno-Economic Analysis (TEA) with key findings now incorporated into the manuscript.

The TEA (Supplementary Figure 13) estimates a Minimum Selling Price (MSP) of \$3.58/kg for bioplastics using an open pond system (500 metric tons/year production), which is competitive. While photobioreactor systems showed higher costs (\$30.68/kg), the open pond scenario is more relevant to the algal productivity and wastewater treatment, highlighting the economic and process feasibility. This minimal selling price is already competitive, the current bioplastics price at about \$4-6 per Kg (Liu et al., 2021). These analyses provide quantitative support for RUMBA's real-world potential and pathways for enhanced economic viability.

Supplementary Figure 13

In the main text, we have added a paragraph to discuss the results of the TEA and the economic feasibility of the RUMBA technology (Line 437-line 459). It reads: “*A techno-*

economic analysis (TEA) was also conducted to evaluate the economic outlook of the RUMBA-based wastewater treatment and bioplastic production, assuming an annual production of 500 metric tons of bioplastics⁴⁵. Economic assumptions and equipment costs are detailed in Supplementary Tables 3-5. The analysis compares the minimum selling price (MSP) of the bioplastic under two algae cultivation systems: an open pond and a photobioreactor, integrating results from the LCA case (Supplementary Figure 13 with byproduct displacement). The analysis indicates an MSP of \$3.58/kg for the open pond system (Supplementary Figure 13). This price falls within the lower end of the reported range of \$3.52 to \$146/kg for bioplastics^{43,45}. Such an MSP is competitive in the bioplastics market as common bioplastics like PHA kg(Polyhydroxyalkanoates), which has an MSP around \$4 to \$6 per kg⁴⁶. The MSP of bioplastics from the open pond system is most sensitive to bioplastic yield and residence time, with material cost being the largest component in the MSP (Supplementary Figures 13 and 14). When using photobioreactor cultivation, the MSP of bioplastic increases to \$30.68/kg, which is not competitive with the current market⁴⁷, primarily due to the higher capital investment required for photobioreactor systems (Supplementary Figure 14). It is worth noting that current PBR systems are not scaled for environmental applications. Open pond system is actually more relevant to the algal bioproduction and wastewater treatment process. A comparison with existing literature suggests this approach has the potential to lower the MSP relative to previous estimates⁴⁸. Furthermore, the current estimate did not account for carbon credits and the environmental benefits of removing microplastics, which could further reduce the MSP of upcycled plastics. Overall, the results indicated that the synergy of microplastics removal, upcycling, and algal bioproduction could lead to economically viable microplastics remediation strategies.”

7. The manuscript could benefit from a more comprehensive explanation of how the microplastic removal efficiency and HCC upcycling performance can be optimized. I recommend further discussion of these factors in the interpretation of the results to provide more insight into potential improvements.

Response: Thank you for this suggestion. We have expanded the **Discussion** section to elaborate on potential strategies for optimizing both microplastic removal efficiency and the upcycling performance of HCC. The section now reads: *“While this study establishes the significant potential of RUMBA, several avenues for improvement are critical for its large-scale implementations. Future work should focus on optimizing the upcycling process for complex environmental scenarios, which often feature low concentrations and diverse mixtures of microplastics and other impurities. For instance, systematic studies are*

required to evaluate how conventional pollutants, such as heavy metals or organic contaminants, might impact HCC viability and the efficiency of the hydrophobic interaction mechanism, in addition to managing factors like COD in broader wastewater applications ^{52,53,54,55}. *Furthermore, a deeper investigation into the roles of specific biopolymers from the cyanobacteria is key to systematically optimizing the composite's mechanical properties, potentially by fine-tuning process parameters like cell-to-plastic ratios or residence times. On the biological front, further synthetic biology efforts could enhance microplastic-HCC interaction, boost removal capacity, and improve overall bioproductivity. Crucially, while the system demonstrated robust performance over the 19-day trial, long-term industrial application necessitates a focus on stability. The effort would involve both process development to maintain optimal system conditions for consistent performance and further strain engineering to ensure genomic stability. The combined efforts can reduce the risk of critical mutations that could compromise the hydrophobic phenotype over extended operational periods. By addressing these challenges, RUMBA can evolve from a promising concept into a versatile and robust platform, effectively integrating microplastic removal with broad environmental remediation and downstream valorization.*” in L501-520.

8. It is suggested to explore applications of the RUMBA system under different environmental and water quality conditions, and discuss whether the presence of conventional pollutants in the wastewater treatment system affects RUMBA’s performance.

Response: We appreciate this insightful comment. We have included experiments using real surface water and wastewater samples (Results section: "HCC removes microplastics at environmentally relevant conditions for water treatment" and "HCC removes environmental microplastics from the surface water and wastewater"). The Discussion section has also been revised to acknowledge that further studies would be beneficial to elaborate on the system's performance under these environmentally relevant conditions. This part reads (Line 502-line 507): *“Future work should focus on optimizing the upcycling process for complex environmental scenarios, which often feature low concentrations and diverse mixtures of microplastics and other impurities. For instance, systematic studies are required to evaluate how conventional pollutants, such as heavy metals or organic contaminants, might impact HCC viability and the efficiency of the hydrophobic interaction mechanism, in addition to managing factors like COD in broader wastewater applications* ^{52,53,54,55}.”

9. Please provide an explanation of the meaning of L524 and ZM in Figure 2.

Response: Our apologies for this oversight. “L524” refers to the engineered strain. “ZM” means “Zoom-in” in the original submission. We have carefully reviewed Figure 2 and revised the labels. The revised Figure 2 is attached below, where “L524” has been replaced with HCC and ZM is spelled as “Zoom in”. Meanwhile, WT is used to denote the wild-type strain.

Figure 2

10. Please carefully check the manuscript for errors, such as the reference to Figure 2h in Line 158, which should be Figure 1h, and the reference to Figure 4 in Lines 213-215, which should be Figure 3. Please also carefully verify and correct the figure numbering in the Supplementary Information to ensure consistency with the main text.

Response: Thank you for spotting these errors. We have meticulously re-checked all figure references throughout the manuscript and corrected the instances you mentioned (Figure 2h now correctly refers to Figure 1f in the revised figure and manuscript in L200, and Figure 4 now correctly refers to Figure 3 in L269-274). We have also thoroughly verified and

corrected the numbering and referencing of figures and tables in the Supplementary Information to ensure complete consistency with the main text.

11. Please added Life Cycle Assessment (LCA) and Techno-economic Analysis (TEA) in your manuscript. The environmental benefits and economic feasibility of RUMBA can be evaluated.

Response: We appreciate the comments and agree that LCA and TEA are crucial for a full evaluation of RUMBA's environmental benefits and economic feasibility. We have conducted additional LCA and TEA analyses. Please check response for comment 5 and 6 and the LCA and TEA section in the main text for details.

12. Was Supplementary Table 1 missed in the supplement?

Response: Thank you for pointing this out. We have double-checked the Supplementary Information. Supplementary Table 1, detailing nutrient removal from wastewater, is now included and correctly referenced in the revised manuscript. The table is also attached below:

Supplementary Table 1. Wastewater Nutrient Removal by Cyanobacteria.

	Nitrate removal (%)	Ammonia removal (%)	Phosphate removal (%)
Influent	47.4 ± 0.6 (b)	100.0 (a)	34.6 ± 1.1 (b)
Effluent	97.5 ± 0.4 (a)	N.A.	37.8 ± 1.9 (b)
Influent + BG11	96.7 ± 0.1 (a)	100.0 (a)	99.8 ± 0.1 (a)
Effluent + BG11	96.4 ± 0.1 (a)	100.0 (a)	99.8 ± 0.1 (a)

Means followed by the same letter are not significantly different. For example, for all the results with (a), they are not different from one another. For results assigned with (a) and (b) separately, the results with (a) are different from the results assigned with (b).

Reviewer 3

The manuscript titled "Remediation and Upcycling of Microplastics by Algae (RUMBA) Synergizing with Wastewater Nutrient Removal and Bioproduction" presents an innovative approach to addressing the critical environmental challenge of microplastic pollution. The authors attempt to integrate synthetic biology methodologies, wastewater treatment, and sustainable upcycling, tackling an issue of significant environmental relevance.

Although the manuscript explores an important and promising topic, it presents several shortcomings that undermine its robustness and scientific credibility. Significant revisions are required, particularly regarding statistical analysis of the data, contextualization within the existing literature, justification of methodological choices, and consistency in data presentation. The manuscript can only be considered for publication after addressing these issues.

Response: we thank the reviewer for the thorough review and constructive feedback on our manuscript and appreciate that the reviewer recognizes our technology as an “an innovative approach to addressing the critical environmental challenge of microplastic pollution. The authors attempt to integrate synthetic biology methodologies, wastewater treatment, and sustainable upcycling, tackling an issue of significant environmental relevance”. We have carefully considered all the comments and have made significant revisions to the manuscript to address the concerns raised.

Comment 1: The statistical treatment of the data must be performed and presented. Vague expressions such as “Substantial higher microplastic removal rate” or “centrifuged suspension were substantially lower in...” lack scientific meaning. The discussion of the results and conclusions must be revised based on statistical analysis. The absence of robust statistical analysis and more detailed data presentation undermines the credibility of the results.

Response: We thank the reviewer for highlighting the need for more rigorous statistical analysis and clearer presentation of our data. We agree that quantitative comparisons supported by statistical tests are essential. We have now performed statistical analyses (e.g., t-tests, ANOVA where appropriate) for all relevant datasets. P-values and significance levels are reported throughout the Results section and in the figure legends. Vague expressions such as “substantially higher” or “substantially lower” have been replaced with precise descriptions supported by these statistical results. The discussion and conclusions have been revised to reflect these statistically validated findings. In addition, a section on Statistical Analysis has been added to the Materials and Methods (Line 711-line 717), as follows:

” All experiments were performed in at least triplicates. Data are presented as mean \pm standard deviation. Statistical significance between groups was determined using a two-tailed Student’s t-test. A p-value of < 0.05 was considered statistically significant. Statistical differences among treatment groups for the integration systems (Table 2) were evaluated

using a one-way Analysis of Variance (ANOVA), followed by a Tukey's HSD post-hoc test to identify significant differences between specific pairs of means ($p < 0.05$).”

Comment 2: The authors must justify the use of genetically modified cyanobacteria instead of naturally occurring species and discuss the implications of this choice. A detailed comparison with existing literature is necessary. A complete characterization of both strains (including the wild type) must also be provided.

Response: We appreciate the reviewer's request for justification and further characterization. We have extensively revised the Introduction to cite state-of-the-art for microplastics removal and justify the use of engineered strain. In summary, we have highlighted the benefits of using bio-based methods among different microplastic remediation technologies. Among different bio-based platforms, we highlighted the need for the engineered strain. To the best of our knowledge, none of the natural strains would achieve the demonstrated performance. The detailed comparison with existing literature is summarized below:

Specifically, in L53-57, we added more details about other microplastic removal strategies *“For example, one study developed core-shell superparamagnetic nanoparticles that attract and glue various microplastics into larger agglomerates, which can then be removed from water with an external magnetic field⁹. Another study employs an interfacial solar evaporation platform to simultaneously produce clean water and significantly improve microplastic removal from the source¹⁰.”*

In 63-L78, we expanded and reframed the background information about existing technology: *“To address the challenges of cost and secondary waste, an ideal solution would integrate microplastic removal with a valorization strategy¹³. This reframes microplastics from an environmental hazard into a potential carbon-rich feedstock for upcycling, a key step toward a circular economy¹⁴. However, a significant technological barrier remains: the high cost and inefficiency of current separation methods make it unfeasible to generate a microplastic stream for conventional recycling¹². A paradigm shift is needed, moving from simple removal to a synergistic system that can both capture microplastics and simultaneously generate value. Here, biological systems offer unique opportunities. If an organism could not only sequester microplastics but also produce a valuable co-product, such as biomass or bioplastics^{15,16,17}, the entire process might become economically viable. This approach would create a direct downstream pathway for the captured plastics, embedding them within the biomass for co-processing and eliminating the risk of re-release into the environment^{18,19}. While algal biomass has been*

explored for bioplastics production^{15, 16, 17}, no current research has proposed synergizing microplastic capture with the simultaneous manufacturing of a valuable algal biomass, a strategy that could finally overcome the economic and logistical hurdles of microplastic remediation.”

In L81-L90, we added more details about previous studies and their limitations: *“Conventional microbial biodegradation methods will depend on the type of plastics to be remediated and have limitations when applied to the environmental remediation where a variety of different types of microplastics exist²⁰. An alternative mechanism involves cell surface adhesion, where recent studies have found a strong positive correlation between a microbe's extracellular polymeric substances (EPS) production and its microplastic removal capacity^{21,22,23}. For example, *Gloeocapsa* sp., which produces high levels of EPS, demonstrated excellent microplastic removal capacity, while poor producers such as *Microcystis panniformis* and *Synechococcus elongatus* PCC 7942 showed minimal MP removal^{21,23}. Furthermore, modulating *Pseudomonas aeruginosa* EPS production has successfully achieved ‘capture and release’ of microplastics^{21,23}.”*

In L103-L111, we added justification about using our new engineered strain and cited the proper previous studies about the detailed engineering design: *“The engineering design was achieved through computational modeling-guided synthetic biology design of high limonene production, and the base strain *Synechococcus elongatus* UTEX 2973 (UTEX 2973) is known for high productivity and absence of microcystin producing genes^{27, 28, 29}. Such hydrophobic-mediated cell-to-cell interactions and self-aggregation inspired us to explore whether hydrophobic effects could also drive interactions between hydrophobic cells and microplastics, given that most plastics are highly hydrophobic. This approach could not only offer an efficient and sustainable method but also introduce a novel mechanism for microplastic remediation.”*

As for the detailed comparison of relevant studies, we presented it in Table 1 and revised it with additional information. Table 1 is included in the main text and can also be found on Page 10 of this response letter.

Based on the comments, we have added more characterization data for the two strains, including limonene data, BATH assay for hydrophobicity, Zeta potential, and EPS assay as suggested by the reviewer. The GC-MS peak for limonene and the EPS measurements are presented in Supplementary Figure 1. The results of the BATH assay and Zeta potential measurements are shown in the revised Figure 1. Some of the new figures are provided below as examples (from supplementary Figure 1 and Figure 1).

GC-MS limonene peak

EPS

BATH assay

Zeta potential

Comment 3: The study addresses a highly relevant environmental issue with potential impact on sustainability and wastewater treatment fields. However, the lack of detailed justifications and a stronger contextualization within the existing literature limits its current scientific contribution.

Response: We thank the reviewer for acknowledging the importance of our work and its potential impact on sustainability and wastewater treatment. We agree that strengthening justifications and contextualization is important. As addressed in our response to Comment 2, we have significantly revised the Introduction section to provide more detailed justifications for our methodological choices and to better contextualize our findings within the existing literature. We have also compared more studies in the Results, Discussion and Methods sections. The total reference number increased to 65 from 35. We believe these changes more clearly highlight the scientific contribution from the field and novelty of the RUMBA system, compared to the state-of-the-art. For example, the relevant edited part in the introduction section now reads (Lines 63-111):

“To address the challenges of cost and secondary waste, an ideal solution would integrate microplastic removal with a valorization strategy¹³. This reframes microplastics from an environmental hazard into a potential carbon-rich feedstock for upcycling, a key step toward a circular economy¹⁴. However, a significant technological barrier remains: the high cost and inefficiency of current separation methods make it unfeasible to

generate a microplastic stream for conventional recycling¹². A paradigm shift is needed, moving from simple removal to a synergistic system that can both capture microplastics and simultaneously generate value. Here, biological systems offer unique opportunities. If an organism could not only sequester microplastics but also produce a valuable co-product, such as biomass or bioplastics^{15,16,17}, the entire process might become economically viable. This approach would create a direct downstream pathway for the captured plastics, embedding them within the biomass for co-processing and eliminating the risk of re-release into the environment^{18, 19}. While algal biomass has been explored for bioplastics production^{15,16,17}, no current research has proposed synergizing microplastic capture with the simultaneous manufacturing of a valuable algal biomass, a strategy that could finally overcome the economic and logistical hurdles of microplastic remediation.

*Among different microplastic remediation technologies, bioremediation is more environmentally friendly because microorganisms are renewable, easy to grow, and the method is compatible with various downstream processes. Conventional microbial biodegradation methods will depend on the type of plastics to be remediated and have limitations when applied to the environmental remediation where a variety of different types of microplastics exist²⁰. An alternative mechanism involves cell surface adhesion, where recent studies have found a strong positive correlation between a microbe's extracellular polymeric substances (EPS) production and its microplastic removal capacity^{21,22,23}. For example, *Gloeocapsa* sp., which produces high levels of EPS, demonstrated excellent microplastic removal capacity, while poor producers such as *Microcystis panniformis* and *Synechococcus elongatus* PCC 7942 showed minimal MP removal^{21,23}. Furthermore, modulating *Pseudomonas aeruginosa* EPS production has successfully achieved 'capture and release' of microplastics^{24,25}. Using microbial EPS to remove microplastics is largely dependent on the species and their capacities vary significantly^{22,23}. Moreover, the EPS-based microplastic removal is relatively slow and inefficient (Table 1)^{21,22,24,25}. For example, it takes *Synechococcus* sp. PCC 7002 over six hours to achieve a modest removal rate (around 80%) of microplastics by the high EPS-producing strain²¹. Furthermore, the utilization of microbial microplastics remediation will always have to take into consideration of toxicity, as the previously mentioned strains like *Microcystis panniformis* are also notorious for producing microcystin to endanger human, livestock and wildlife health².*

Despite the limitations, microbial microplastics removal could achieve economic and sustainable process if the organism can produce useful bioproducts, remediate other contaminants, and achieve high efficiency of microplastic removal. In this regard, cyanobacteria have the potential to synergize the excess nutrient utilization. In a previous work, it is showed that engineering a hydrophobic cell surface can promote

*cyanobacterial self-aggregation, thus facilitating the low-cost recovery of the biomass at a high efficiency(*Long et al., 2022). *The engineering design was achieved through computational modeling-guided synthetic biology design of high limonene production, and the base strain Synechococcus elongatus UTEX 2973 (UTEX 2973) is known for high productivity and absence of microcystin producing genes* ^{27, 28,29}. *Such hydrophobic-mediated cell-to-cell interactions and self-aggregation inspired us to explore whether hydrophobic effects could also drive interactions between hydrophobic cells and microplastics, given that most plastics are highly hydrophobic. This approach could not only offer an efficient and sustainable method but also introduce a novel mechanism for microplastic remediation.”*

Comment 4: The authors must justify the selected concentrations and sizes of the microplastics used, considering the characteristics of the water samples studied, and provide robust support for the experimental and analytical conditions employed.

Response: We appreciate this point and have clarified our choices. We have selected the size based on previous publications(Reddy and Nair, 2022; Wang et al., 2023). To justify the selection of the size and concentration of the microplastics used, we have modified the manuscript accordingly and inserted the following sentences in lines 547 to 553.

“The selection of these specific sizes was based on several considerations. The chosen sizes reflect the general distribution observed in wastewater treatment plants and acknowledge the inherent difficulty in removing smaller, nano-sized particles^{11,58}. To evaluate the efficacy of hydrophobicity-mediated aggregation and sedimentation across different particle sizes, PS microplastics (200, 500, and 800 nm) were spiked into water samples. The final concentration of 0.02% (or as otherwise specified) is consistent with previous studies ^{9, 59} and allowed for a robust test of the removal mechanism with particle sizes relevant to environmental samples.”

Particularly in our study, different microplastic concentrations were used or tested for different purposes. For example, in the experiments shown in Figure 1, gradient concentrations of 200 nm polystyrene microplastics were mixed with WT and HCC cells at final concentrations of 0%, 0.001%, 0.005%, 0.01%, 0.02%, and 0.05%. The gradient design demonstrated the different sedimentation levels between treatments, which clearly indicate the potential interactions between HCC and microplastics (Figure 1b-c). In fact, the concentration we selected is commonly used in microplastic removal studies. One example is the study by Sarcletti et al., “The remediation of nano-/microplastics from water”, published in Materials Today(Sarcletti et al., 2021). The authors used 0.1% wt%

concentrations for their characterization. Other recent studies use similar concentrations. The study by Shen et al used “0.05, 0.1, 0.2, 0.5, 0.8, and 1 g L⁻¹” in microplastics removal using an electrocoagulation process (Shen et al., 2022). Our microplastics concentration selection is thus similar to the published work and follows the practice standards. The two references are cited in our revision (lines 551-553), which read: *“The final concentration of 0.02% (or as otherwise specified) is consistent with previous studies^{9,59} and allowed for a robust test of the removal mechanism with particle sizes relevant to environmental samples.”*

Comment 5: Basing microplastic sedimentation on the hydrophobicity of limonene in HCC requires confirmation of the absence of EPS production and an analysis of the charges of the cells and microplastics. These data are crucial and must be provided.

Response: We appreciate the reviewer’s comments and agree that these data are crucial for supporting the proposed mechanism. We have carried out additional experiments to measure the EPS and cell charge. The results are included in the revised manuscript, specifically:

In L231-235: *“Furthermore, measurements of extracellular polysaccharides, the major components of extracellular polymeric substances (EPS), revealed that WT cells produced a higher level of extracellular polysaccharides compared to HCC cells (Supplementary Figure 1g). This finding indicates that the observed cell-microplastic interaction in HCC is likely independent of EPS production, further supporting the role of limonene-induced surface hydrophobicity.”*

Supplementary Figure 1g. Extracellular polysaccharides measurements.

And in L248 -260: *“Furthermore, the zeta potential measurements were performed to further elucidate the interaction mechanisms between HCC and microplastics. The PS microplastics exhibited consistently negative zeta potentials: -52.42 ± -0.26 mV, -56.16 ± -0.73 mV, and -63.88 ± -0.69 mV for 200 nm, 500 nm, and 800 nm sizes, respectively (Figure 1h). In contrast, HCC cells demonstrated a significantly ($p < 0.01$) lower zeta*

potential of -13.91 ± -1.47 mV, compared to -25.78 ± -0.69 mV for WT cells (Figure 1h). This reduction in the negative surface charge on HCC cells can be attributed to their increased surface hydrophobicity, which likely lessens the adsorption of polar molecules and charged ions. This, in turn, leads to a reduced negative charge in the stern layer of HCC cells and a diminished overall zeta potential. Taken together, these results highlight the dual roles of increased cell hydrophobicity in mediating HCC-PS interactions: it weakens electrostatic repulsion by decreasing the cell's negative charge, while simultaneously enhancing cell-to-microplastic attraction through increased hydrophobic interactions.”

Figure 1h. Zeta Potential

Comment 6: The methodology shows significant gaps. The lack of justification for the experimental conditions and tested parameters undermines the study's robustness. Additionally, the described methods do not provide sufficient detail to ensure the reproducibility of the work.

Response: We appreciate the reviewer's comments on the methodology. We have thoroughly reviewed and revised the entire Methods section. We have added further justifications for specific experimental conditions and parameters. We have also expanded on procedural details where necessary to ensure that other researchers can reproduce our work. These include a more detailed description of strain and growth conditions, microplastic removal assay, suspension analysis, and addition of Zeta potential, TEA & LCA analyses, and Statistical analysis. The revised sections are listed below:

Strains and growth condition

*The wild type *Synechococcus elongatus* UTEX 2973⁵⁶ was generously gifted by Dr. Himadri B. Pakrasi from Washington University in St. Louis, where extensive genomics information is available to reveal the rapid growth mechanism⁵⁷. The limonene-producing HCC strain was developed as previously described, where a synthetic promoter has driven the ultra-high level limonene production, leading to hydrophobic cell surface and auto-*

sedimentation^{27,28}. Both strains were maintained in 250-mL Erlenmeyer flasks containing 50 ml of BG11 medium (Sigma, C3061) supplemented with 10 mM TES buffer (pH 8.2) and grown under a photon flux density of 50 $\mu\text{mol photons m}^{-2} \text{ s}^{-1}$ at 30°C. For large-scale cultivation, Strains were cultivated in 1-L Roux bottle containing 500 ml of BG11 medium. These cultures were maintained at 37°C and sparged with 5% (v/v) CO₂ in air at a flow rate of approximately 0.8 L/min. Fed-batch cultivation was performed every 24 h with 1× BG11, unless otherwise specified, starting from the second day of cultivation. This process continued until the OD₇₃₀ reached 15, where the late log phase is often also used for algal harvest. This concentration thus could be used to synergize algal bioproduction, microplastic removal, and downstream upcycling. The concentration also ensures a high cell density while maintaining strong aggregation capacity. The temperature was maintained at 37 °C throughout the cultivation, and the light regime was set to one-side 357 $\mu\text{mol m}^{-2} \text{ s}^{-1}$, one-side 574 $\mu\text{mol m}^{-2} \text{ s}^{-1}$, and double-side 574 $\mu\text{mol m}^{-2} \text{ s}^{-1}$ for 0 - 12 h, 12 - 36 h, and thereafter.

Microplastic removal assay

The following microplastics were used in this study: polystyrene (PS) nanoparticles of 200 nm (Cat# 69057), 500 nm (Cat# 59769), and 800 nm (Cat# 65984) from Sigma-Aldrich; polyethylene terephthalate (PET) particles (<300 μm , Cat# 1000080955) from Goodfellow; and polyethylene (PE) particles (32-38 μm , Cat# DNP-RPE03) from CD Bioparticles. The PS particles were supplied as aqueous suspensions. The PET and PE particles were supplied as powders, which were weighed on an analytical balance (Sartorius) before being added directly to treatment samples. The selection of these specific sizes was based on several considerations. The chosen sizes reflect the general distribution observed in wastewater treatment plants and acknowledge the inherent difficulty in removing smaller, nano-sized particles^{11, 58}. To evaluate the efficacy of hydrophobicity-mediated aggregation and sedimentation across different particle sizes, PS microplastics (200, 500, and 800 nm) were spiked into water samples. The final concentration of 0.02% (or as otherwise specified) is consistent with previous studies^{9,59} and allowed for a robust test of the removal mechanism with particle sizes relevant to environmental samples.

Cyanobacterial cultures were harvested at an OD₇₃₀ of approximately 15. For removal assays, cells were mixed with PS microplastics to the designated final concentrations. For 200 nm PS, experiments were conducted in a total volume of 20 mL. For 500 nm and 800 nm PS, a total volume of 6 mL was used. The mixtures were gently inverted about 10 times to ensure homogeneity and then allowed to settle undisturbed for 1 hour at room temperature. To evaluate the microplastic removal in wastewater and surface water environments, harvested cyanobacterial cells were gently concentrated by low-speed centrifugation (1000

×g for 10 minutes). The resulting pellet was then gently resuspended in the wastewater or surface water samples spiked with commercial or environmentally sourced microplastics.

The interaction between cyanobacteria and various microplastics was qualitatively assessed using fluorescence and brightfield microscopy. Wild-type or HCC cells from cultures at an OD_{730} of ~15 were mixed with either PET, PE, PS (10 μm , Sigma 61946), or microplastics isolated from environmental water samples. The mixtures were mixed gently by inversion to facilitate interaction and incubated for about 30 minutes to allow aggregation. Following incubation, a 10 μL aliquot was carefully withdrawn from the bottom of the tube using a wide-bore (cut-off) pipetted onto a glass slide, and covered with a coverslip. For commercial microplastics (PET, PE, and PS), samples were imaged using a Leica DM6B fluorescence microscope. Cyanobacterial cells were visualized by their characteristic red chlorophyll autofluorescence using a Cy5 filter set. The intrinsic blue-cyan autofluorescence of PET and PE microplastics was observed using a DAPI filter set. For environmental samples, interactions were observed using a Boreal bright-field microscope (Boreal Optical, China) equipped with a Moticam X5 Plus camera (Motic, Richmond, BC, Canada).

The long-term stability of microplastic removal was evaluated in a semi-continuous fed-batch system. The initial culture was established in a 1-L Roux bottle containing 500 mL of 2× BG11 medium, spiked with 0.05 g/L of 800 nm PS microplastics, and fed daily with 1× BG11 medium. Cultivation conditions were identical to those described above. After an initial 4-day cultivation period, 50 mL aliquots ($n=3$) were aseptically collected for analysis. The remaining culture was then used to inoculate the subsequent batch. To begin the next cycle, the OD_{730} of the remaining culture was measured to calculate the required inoculum volume. This volume was transferred to a new Roux bottle containing fresh 2× BG11 medium to achieve a starting OD_{730} of approximately 2.0. The new culture was re-spiked with 0.05 g/L of 800 nm PS microplastics. This reset and cultivation process was repeated, with subsequent sampling occurring every three days up to a total of 19 days. A similar protocol was applied using wastewater influent or effluent as the growth medium to assess the integration of microplastic removal with wastewater treatment.

Suspension analysis for PS microplastic removal by cyanobacteria

After settling, aliquots from the top of the suspension were analyzed. Turbidity and chlorophyll fluorescence were measured in 200 μL samples in a 96-well plate using a SpectraMax iD5 microplate reader (Molecular Devices). Turbidity was measured by absorbance at 850 nm (a standard wavelength for turbidity sensors due to its minimized color interference(Chu et al., 2023), and chlorophyll fluorescence was measured with an excitation wavelength of 595 nm and an emission wavelength of 690 nm, which was

optimized by the a SpectraMax iD5 microplate reader. All samples were diluted with BG11 medium as needed to maintain absorbance readings within the linear range of the instrument (0.3-0.7).

Differential centrifugation was applied to separate cyanobacterial cells and PS microplastics and to estimate the remaining PS microplastics in the suspension. Centrifugation speeds were set to 800 ×g for samples containing 200 nm microplastics and 300 ×g for 500 and 800 nm microplastics, and the centrifugation time was set to 3 minutes. The centrifugation procedures were effective in achieving good separation (Supplementary Figure 3), which validated the reliability of the protocol. The relative turbidity (850 nm) of PS microplastic was calculated by normalizing the turbidity of HCC samples to that of WT control samples to account for any cell pelleting during centrifugation.

Zeta Potential Measurement

The zeta potentials of cyanobacterial cells and PS microplastics were measured using a Zetasizer Advance Pro (Malvern Panalytical). For the analysis, aliquots of both cell and microplastic suspensions were diluted with ddH₂O to a suitable concentration and maintained at room temperature. To minimize potential changes in cell surface hydrophobicity following harvesting, cyanobacterial samples were analyzed within 3 hours of removal from the growth culture.

Techno-Economic and Life Cycle Analysis (TEA/LCA)

TEA and LCA were performed to evaluate the economic feasibility and environmental impact of the RUMBA process. The detailed methodology, system boundary, and assumptions for these analyses are provided in Supplementary Information (Supplementary Figures 11&14, Supplementary Tables 2-5). The scenario and sensitivity analyses, which identify the key factors that impact the environmental and economic performance of RUMBA process, are shown in Supplementary Figures 12-13, Tables 6-7).

Statistical Analysis

All experiments were performed in at least triplicate. Data are presented as mean ± standard deviation. Statistical significance between groups was determined using a two-tailed Student's t-test. A p-value of < 0.05 was considered statistically significant. Statistical differences among treatment groups for the integration systems (Table 2) were evaluated using a one-way Analysis of Variance (ANOVA), followed by a Tukey's HSD post-hoc test to identify significant differences between specific pairs of means (p < 0.05).

Comment 7: The entire manuscript requires a thorough review, as it contains numerous errors, including issues with figure numbering, language inconsistencies, and methodological gaps.

Response: We thank the reviewer for pointing this out and apologize for the errors. We have conducted a thorough review of the entire manuscript and supplementary materials. We have corrected all identified figure numbering issues, ensured consistency in language and terminology (e.g., acronym usage), and addressed methodological gaps as detailed in other responses. We have also performed a careful proofread to correct grammatical errors and improve overall clarity.

Comment 8: (Line 30 and others) Use impersonal language throughout the manuscript and supplementary materials.

We thank the reviewer's recommendation in using impersonal language. This is actually our preference. However, we also acknowledge that other description styles are commonly accepted by the community nowadays. A recent example can be seen in this recently published paper by Nature, "Identification and characterisation of vaginal bacteria-glycan interactions implicated in reproductive tract health and pregnancy outcomes, published June 5, 2025". We thus followed the Reviewer's comments and adopted impersonal language, wherever it was necessary throughout the manuscript. For some occasions, we kept the expressive language to keep the flow.

Comment 9: (Line 32) Consider replacing "new technology" with "system."

Response: We thank the reviewer for this suggestion. We have replaced "new technology" with "system".

Comment 10: (Line 78) Provide the zeta potential of the cells and microplastics.

Response: We thank the reviewer for this suggestion. We have carried out additional experiments and included zeta potential in the revised manuscript (L248-260). The Figure and results are attached below: *"Furthermore, the zeta potential measurements were performed to further elucidate the interaction mechanisms between HCC and microplastics. The PS microplastics exhibited consistently negative zeta potentials: -52.42 ± -0.26 mV, -56.16 ± -0.73 mV, and -63.88 ± -0.69 mV for 200 nm, 500 nm, and 800 nm sizes, respectively (Figure 1h). In contrast, HCC cells demonstrated a significantly (p*

< 0.01) lower zeta potential of -13.91 ± -1.47 mV, compared to -25.78 ± -0.69 mV for WT cells (Figure 1h). This reduction in the negative surface charge on HCC cells can be attributed to their increased surface hydrophobicity, which likely lessens the adsorption of polar molecules and charged ions. This, in turn, leads to a reduced negative charge in the stern layer of HCC cells and a diminished overall zeta potential. Taken together, these results highlight the dual roles of increased cell hydrophobicity in mediating HCC-PS interactions: it weakens electrostatic repulsion by decreasing the cell's negative charge, while simultaneously enhancing cell-to-microplastic attraction through increased hydrophobic interactions.

Figure 1h. Zeta Potential

Comment 11: (Line 82) Improve the state of the art. Explain how limonene facilitates the self-aggregation of microplastics. Is limonene present on the surface of the cells? Does the cyanobacterium used not produce EPS? Are there natural species that produce limonene? What is the WT (Wild Type)? Provide a complete characterization of both strains, including the WT.

Response: We appreciate these questions regarding the improvement over the state-of-the-art. First, we have expanded the citation and description regarding how the strain engineering was carried out for limonene production. There are no reported natural cyanobacteria species producing limonene. The WT is *Synechococcus elongatus* UTEX 2973. The description of limonene engineering and WT was described in both L103-L111 and L523-L514.

In L101-L109: “The engineering design was achieved through computational modeling-guided synthetic biology design of high limonene production, and the base strain *Synechococcus elongatus* UTEX 2973 (UTEX 2973) is known for high productivity and absence of microcystin producing genes^{28, 27, 29}. Such hydrophobic-mediated cell-to-cell interactions and self-aggregation inspired us to explore whether hydrophobic effects

could also drive interactions between hydrophobic cells and microplastics, given that most plastics are highly hydrophobic. This approach could not only offer an efficient and sustainable method but also introduce a novel mechanism for microplastic remediation.”

In L523-L527: *“The wild type *Synechococcus elongatus* UTEX 2973⁵⁶ was generously gifted by Dr. Himadri B. Pakrasi from Washington University in St. Louis, where extensive genomics information is available to reveal the rapid growth mechanism⁵⁷. The limonene-producing HCC strain was developed as previously described, where a synthetic promoter has driven the ultra-high level limonene production, leading to hydrophobic cell surface and auto-sedimentation^{27,28}.”*

Furthermore, we have added sentences to explain the underlying mechanisms of how limonene promotes cell-to-cell aggregation and strain characterization in L132-141. The sentences read: *“The produced limonene was found to be secreted and enriched on the cell surface before volatilizing, thereby increasing cell hydrophobicity²⁷. The unique smooth cell surface of UTEX 2973, resulting from defects in pilus biogenesis, exposes the hydrophobic surface, which enables interactions that promote cell aggregation (Figure 1a)²⁷. Indeed, cell aggregation and self-sedimentation were observed in HCC cells, but not in wild-type (WT) cells (Figure 1b, c, d, e and f). This behavior is linked to limonene production in the HCC samples (Supplementary Figure 1a-d), which increases cell surface hydrophobicity. A subsequent BATH assay confirmed this enhanced hydrophobicity, demonstrating that a larger portion of HCC cells attached to a hydrophobic hexadecane layer compared to WT cells (Figure 1g, and supplementary Figure 1e). These results corroborate with our previous results²⁷ and zeta potential data (Figure 1h)”.*

Moreover, we have compared the EPS production in both WT and HCC. The results show that WT produces more EPS than HCC. Since HCC has much stronger capacity to remove microplastics, the microplastics removal is thereby not EPS dependent. The results were presented in L231-235: *“Furthermore, measurements of extracellular polysaccharides, the major components of EPS, revealed that WT cells produced a higher level of extracellular polysaccharides compared to HCC cells (Supplementary Figure 1g). This finding indicates that the observed cell-microplastic interaction in HCC is likely independent of EPS production, further supporting the role of limonene-induced surface hydrophobicity”.*

Furthermore, we have carried out additional experiments to compare the zeta potential, hydrophobicity, and limonene production between WT and HCC, as detailed in response to comment 2, 5, and 10.

Comment 11: (Line 117) Perform statistical analysis to support the conclusions.

Response: We appreciate the reviewer's comments. Statistical analyses have been performed for the Supplementary Figure 1b (originally Figure 1c). The statistical results are included in the figure legend. The Figure legend reads: *“(b) Suspension turbidity of HCC-PS samples was significantly lower ($p < 0.01$) than that of WT-PS samples across all tested PS concentrations.”* in L24-L26 in supplementary information.

Comment 12: (Line 119-123) Since turbidity and fluorescence depend on the species, these analyses should be normalized using a control.

Response: We agree that controls are needed when measuring turbidity and fluorescence. However, we also want to point out that the strains used in this study belong to the same species, with the only difference being that one strain is engineered to produce limonene while the other is not. These two strains are expected to be very similar in terms of turbidity and fluorescence characteristics. Thus, the WT strain serves as a reliable control, better than any other cyanobacterial species. The significant differences observed in Figures 1c and 1d at the 0% polystyrene microplastics concentration do not arise from inherent differences in turbidity or fluorescence between the strains. Instead, these differences result from cell aggregation and sedimentation in the HCC strain, even in the absence of polystyrene microplastics, leading to reduced cell concentration in the suspension, as also shown in the image in Figure 1b. Absolute values were used for calculations in Figures 1c and 1d, rather than normalized percentages, to reflect the cyanobacterial response to varying concentrations of polystyrene and to clearly demonstrate the differences between HCC and WT strains.

Comment 13: (Line 126) Provide the density values for microplastics of different sizes.

Response: The density values for microplastics are now provided in the manuscript. All microplastics, regardless of size, have a density of 1.005 g/mL, as the density depends on the material itself but not the size. However, larger diameter microplastics are more prone to settling despite having the same density. Therefore, a lower centrifugation speed is necessary to effectively separate large microplastics from cyanobacterial cells.

The revised sentence containing the density information reads: *“Since the size and density of microplastics (1.005 g/mL) and cyanobacteria cells are different, differential centrifugation thus pelletizes the cell first and leaves most microplastics in the suspension at the low centrifugation speed.”* in L165-L168.

Comment 14: (Line 146-149) Explain the statistical significance of the observed differences.

Response: We appreciate the reviewer's comments. We have updated the statistical analysis between the comparison of HCC sediment fraction and WT sediment fraction (now Figure 1d in the revised version) in the figure legend. The description reads: *“(d) Quantification of sediment and suspension fractions in WT-PS and HCC-PS samples. Sediment fractions in HCC samples were found to be significantly ($p < 0.01$) higher compared to WT samples.”*

Comment 15: (Line 149) Present TGA results with replicates and statistical analysis.

Response: We thank the reviewer for their valuable feedback regarding our TGA data. For Supplementary Figure 15, our primary aim was to illustrate the overall thermal behavior and allow for a direct comparison of the degradation patterns between samples. We were concerned that adding error bars to each full curve within the same plot would create a visually complex figure, potentially obscuring the key comparative message. However, we completely agree that a quantitative representation of the data's variability is crucial. We have included Figure 1g, which is derived from the TGA results, with error bars to represent replicates, and added statistical analysis for this Figure. The revised figure and its updated legend are provided below.

Figure 1g. Polystyrene and biomass composition in suspension fractions. Significantly higher PS fractions ($p < 0.01$) were observed in WT-PS suspensions (65.3%) compared to HCC-PS suspensions (37.8%), whereas HCC-PS suspensions showed higher biomass fractions.

Comment 16: (Line 158) Review figure numbering. Justify the use of 5 mg of microplastics and indicate whether these concentrations reflect environmental conditions.

Response: The figure number has been meticulously reviewed and corrected throughout the manuscript.

The use of 5 mg of microplastics in this specific experiment (Figure 1h) was to determine the removal capacity of the HCC biomass (mg microplastic per g biomass). This requires a sufficient, measurable quantity of microplastics, but also within the removal capacity of HCC cells for a mechanism study. The sentence is revised and now reads: *“Furthermore, the microplastic removal capacity was evaluated by measuring the cell biomass required to remove 5 mg of 200 nm PS microplastics, an amount that is measurable and also within the removal capacity by HCC cells.”* In L198-L200.

Comment 17: (Line 200) Microplastics often adhere to glass walls or aggregate among themselves and sediment. To avoid this, Tween 20 is added for dispersion. Does this suggest that hydrophobicity is the driving force?

Response: Yes, we completely agree with the reviewer’s inference. Tween 20 is selected to support our hypothesis that hydrophobicity is the driving force. The hydrophobic interaction disruption experiment was carried out using Tween 20, a surfactant, which demonstrated that disrupting hydrophobic interactions significantly impairs the microplastic removal efficiency of HCC. This provides functional evidence for the critical role of hydrophobicity in the aggregation and co-sedimentation process. The results were provided in Supplementary Figure 7.

Comment 18: (Line 222-223) Confirm the GPS coordinates.

Response: We appreciate the reviewer pointing out the coordinate error. We have updated it in the manuscript. The section now reads (Line 282-Line 285): *“To do so, water samples from different locations were collected, including surface water samples from a lake in the research park (30.602603, -96.360686) on the Texas A&M University campus, as well as wastewater samples from the Texas A&M wastewater treatment plant (30.564525, -96.370141).”*

Comment 19: (Line 252) Review figure numbering.

Response: We thank this reviewer for careful reading of our manuscript. We have double-checked our figure numbers and corrected them.

Comment 20: (Line 262) explain the term “environmentally relevant conditions.”

Response: We have defined "environmentally relevant conditions" more explicitly in the text (L279-L281, revised). In this context, it refers to using natural surface water and wastewater samples (with their inherent complex matrices, ionic strengths, and presence of natural organic matter) and testing with both spiked model microplastics and enriched environmental microplastics. The revised sentence now reads: *“Given the widespread presence of microplastics in almost all natural water systems³², the performance of the hydrophobic-mediated microplastic removal under environmentally relevant conditions was further evaluated, using natural surface water and wastewater samples”*.

Comment 20: (Line 279) Critically justify the tested concentrations and whether they are environmentally relevant.

Response: The concentrations mentioned in this paragraph (originally L266-278) aim to evaluate the toxicity effects of microplastics on cyanobacteria. Thus a wide range of concentrations were applied to test the boundary of the toxicity effects. The high concentrations are not necessarily related to environmentally relevant concentrations. However, the absence of toxicity at high concentrations suggests that it is unlikely that microplastics will have any negative impact on the algal growth at lower environmentally relevant concentrations. This also suggested that it is possible to synergize algal bioproduction with microplastics removal.

Comment 21: (Line 324) Evaluate the sustainability of the process used.

Response: We appreciate the reviewer’s comments. Based on the reviewer’s comment, we have carried out the LCA analysis for sustainability. The results section in L412-L436 is attached below:

“The techno-economic Analysis (TEA) and life cycle analysis (LCA) were carried out to evaluate the feasibility and scenario for commercial application of RUMBA, along with its emission impacts (Supplementary Table 2-7 and Supplementary Figure 11-14). LCA was performed to evaluate the environmental impact of the RUMBA process integrating with algae cultivation, microplastic removal, wastewater treatment, and bioplastic production. The functional unit, system boundary, and inventory analysis are detailed in the Supplementary Figure 11 and Supplementary Table 2. If the system is powered by conventional energy, the total emission ranges from 19.38 to 20.67 kg CO₂ equivalent per 1 kg of bioplastics, with electricity consumption remaining to be the primary contributor to CO₂ emissions across the RUMBA-based wastewater treatment and bioplastic production life cycle (Scenarios 3 and 4 in Supplementary Figure 12). To promote a circular economy for algae, we also consider the use of renewable energy to power the system⁴¹, which is very feasible as the algal production can be readily coupled with solar

energy as both need sunlight. Without considering residual biomass allocation, the production of 1 kg of bioplastics through upcycling results in a net utilization of 3.21 kg CO₂ emissions using renewable power sources. The actual industrial implementation will certainly carry out the byproducts displacement, and the scenario analysis highlights the potential for this process to achieve negative emissions (-4.50 kg CO₂ emissions/kg of bioplastic produced when considering residual biomass for electricity generation). This stands in contrast to other bioplastic production methods, which typically have positive CO₂ emissions ranging from 0.6 to 282.6 kg CO₂ emissions/kg of bioplastic produced^{42, 43}. In addition to impacts on greenhouse gas emissions, this process confers positive environmental impacts by utilizing nitrogen and phosphate from wastewater and removing microplastics⁴⁴. The results highlighted that the synergy of microplastics removal and upcycling, algal production, and wastewater nutrient usage could achieve a significant positive environmental impact, as shown in Supplementary Table 7.”

Comment 22: (Line 406) Justify why an OD730 of 15 was used. Does this value represent the stationary phase of the species?

Response: We appreciate the question. The OD growth curve is as shown in the Figure below, which has been published in our previous paper(Long et al., 2022). As shown in the figure, OD 730 corresponds to the late-logarithmic phase in our growth setup. Od 15 is often the concentration to harvest algae in a fed batch (FB—orange line in the cited figure) algal production system. Thereby, when microplastic removal is synergized with the algal production, this will be the concentration relevant to commercial application. The justification was added in L534 to L537:

“This process continued until the OD730 reached 15, where the late log phase is often also used for algal harvest²⁷. This concentration thus could be used to synergize algal bioproduction, microplastic removal, and downstream upcycling. The concentration also ensures a high cell density while maintaining strong aggregation capacity.”

Long et. al., *Nature Communications* 13 (1), 541, 2022

Comment 23: (Line 418-419) Justify the choice of wavelengths used.

Response: We appreciate the comments and added relevant references to justify the wavelength usage. 850 nm is a commonly used wavelength for turbidity measurements and is widely used in many turbidity sensors because it is less affected by sample color. Ex595nm and Em690 nm are in-house optimized for chlorophyll measurements with the instrument used. We have included clarifications in the method section. The section is as follow(Line 591-594): *“Turbidity was measured by absorbance at 850 nm (a standard wavelength for turbidity sensors due to its minimized color interference⁶⁰), and chlorophyll fluorescence was measured with an excitation wavelength of 595 nm and an emission wavelength of 690 nm, which was optimized by the a SpectraMax iD5 microplate reader.”*

Comment 24: (Line 426): Explain how the authors account for the turbidity effect of cyanobacteria.

Response: We appreciate the question and have clarified this in the Methods section (Lines 590-592, revised). The turbidity effect of cyanobacteria is accounted for in several ways:

1. Control Samples: WT cyanobacteria (without microplastics or with minimal interaction) serve as controls. The turbidity of these samples provides a baseline.
2. Differential Centrifugation: For quantifying microplastics remaining in suspension, a low-speed differential centrifugation step is employed. This step is optimized (validated in Supplementary Figure 3) to pellet the majority of cyanobacterial cells while leaving most of the microplastics (especially smaller ones) in the supernatant. The turbidity of this cell-depleted supernatant is then measured to estimate the remaining microplastic concentration.
3. Relative Calculations: Microplastic removal efficiency is often calculated relative to the WT control, which inherently normalizes for the behavior of cells alone. A sentence of *“The relative turbidity (850 nm) of PS microplastic was calculated by normalizing the turbidity of HCC samples to that of WT control samples to account for any cell pelleting during centrifugation.”* Was added to the Method (Line 602-604) section for clarification.

Comment 25: (Line 523): Add a section on statistical data analysis.

Response: We appreciate the reviewer’s suggestion. We have added a section of **“Statistical Analysis”** in the Material and Methods section and the paragraph reads (Line 711-line 717): *“All experiments were performed in at least triplicate. Data are presented as mean \pm standard deviation. Statistical significance between groups was determined using a*

two-tailed Student's t-test. A p-value of < 0.05 was considered statistically significant. Statistical differences among treatment groups for the integration systems (Table 2) were evaluated using a one-way Analysis of Variance (ANOVA), followed by a Tukey's HSD post-hoc test to identify significant differences between specific pairs of means ($p < 0.05$)."

Comment 26: (Fig. 2): Review the nomenclature given to the samples. Ensure consistency in the nomenclature throughout the manuscript and supplementary materials.

Response: We appreciate the reviewer for highlighting the inconsistency in the nomenclature. As suggested, we have thoroughly reviewed the manuscript and supplementary materials to ensure that all nomenclature is consistent.

Comment 27: (L689): Replace “polystyrene microplastics” with its acronym. Review all similar cases in the manuscript.

Response: We appreciate the reviewer's suggestions on the acronym. As such, polystyrene has been replaced with PS. We have also thoroughly reviewed the manuscript and supplementary materials to ensure that all acronyms are correctly used.

Comment 28: (Fig. 5): Review the nomenclature given. Ensure consistency throughout the manuscript.

Response: We appreciate the reviewer for highlighting the inconsistency in the nomenclature. As such, “WT” has been replaced by “WT-PS”, “HCC” has been replaced by “HCC-PS”. Meanwhile, we have thoroughly reviewed the figures in the manuscript and supplementary materials to ensure that all nomenclature is consistent.

Comment 29: (Table 1): Add information to the main rows for clarity. Complete the missing references.

Response: We appreciate the review's suggestions and point out the missing reference in the table. We have added additional information to the table to enhance its clarity and added the missing reference. The new table is attached below:

Table 1. Summary of microorganism-based microplastic capture.

Species	Mechanism	Design	Microplastic type and size	Temperature	Efficiency	Ref.
M. panniformis	EPS	N.A.	PS (<106 μm) and PMMA (<250 μm)	20 \pm 1 $^{\circ}\text{C}$	N.A.	23
Scenedesmus sp.						
Tetraselmis sp.						
Gloeocapsa sp.						
Cyanothece sp.	EPS	N.A.	PS (0.1 and 10 μm)	25 \pm 1 $^{\circ}\text{C}$	N.A.	22
Synechococcus PCC 7942	EPS	N.A.	PS (0.1 and 10 μm)	30 $^{\circ}\text{C}$	~18% in 6.5h [#]	21
Synechococcus PCC 7002					~ 82% in 6.5h [#]	
P. aeruginosa	EPS	Knockout of wspF & inducible expression of yhh	PET, PMMA, nylon, PVC (< 106 μm), Microplastics from seawater (106 - 300 μm)	25 $^{\circ}\text{C}$ or 30 $^{\circ}\text{C}$	> 90% in 24h [#]	24
P. aeruginosa	EPS	Expression of trypsin & laboratory evolution	PS, PET, PMMA (< 106 μm), Microplastics from seawater (106 - 300 μm)	37 $^{\circ}\text{C}$	N.A.	25
Synechococcus UTEX 2973	Hydrophobicity	Expression of limonene synthase	PS (<5 μm), PET (<300 μm), PE (35 μm), Microplastics from surface water and wastewater.	37 $^{\circ}\text{C}$ and room temperature	91.4% in 1h	This study

N.A., not applicable or not indicated in the original paper; EPS, extracellular polymeric substances; PS, polystyrene; PMMA, polymethyl methacrylate; PET, polyethylene terephthalate; PVC, polyvinyl chloride; PE, polyethylene.

Comment 30: (Table 2): To enhance clarity, add “Nitrate removal (%), microplastic removal (%), and biomass production (%)” to the table. Provide statistical analysis. Explain what the “integrated system” refers to.

Response: We appreciate the reviewer’s suggestions. We have revised Table 2 and the new table is attached below. The integrated system was referring to the growing algal culture that interacts with microplastics and consumes the nutrients in the water. As it is not clear, we changed it to “ algal cultivation system that removes microplastics and nutrients” (Line 1046-line 1047).

Table 2. Nitrate removal, microplastic removal, and biomass production in the “ algal cultivation system that removes microplastics and nutrients.

Treatment	Nitrate removal (%)	Microplastic removal (%)	Biomass production (g/L)
Influent- day 5	98.7 ± 1.0 (a)	35.8 ± 9.1 (c)	2.46 ± 0.28 (b)
Effluent- day 5	99.4 ± 0.2 (a)	54.3 ± 7.9 (b)	2.54 ± 0.28 (b)
Influent- day 8*	97.6 ± 0.1 (b)	78.5 ± 6.4 (a)	3.80 ± 0.01 (a)
Effluent- day 8*	98.6 ± 0.3 (a)	88.6 ± 1.0 (a)	3.50 ± 0.10 (a)
Influent- day 9*	98.3 ± 0.1 (a)	51.2 ± 1.1 (b)	3.57 ± 0.20 (a)
Effluent- day 9*	99.2 ± 0.4 (a)	65.1 ± 12.8 (b)	3.84 ± 0.19 (a)

Means followed by the same letter are not significantly different. For example, for all the results with (a), they are not different from one another. For results assigned with (a) and (b) separately, the results with (a) are different from the results assigned with (b).

* Additional nutrients (equivalent to 1× BG11) were fed after day 5.

Comment 31: (Supplementary Figures): Revise the formatting to match the journal's standards and ensure correct citation in the text.

Response: We appreciate the reviewer’s suggestions on the supplementary data. We have redone the formatting of the supplementary data and thoroughly reviewed the citation of supplementary results in the main text.

Comment 32-33-34: (Supplementary Figure 1): Provide statistical analysis between series and time points; (Supplementary Line 8): Review to avoid repeating units in consecutive numbers; (Supplementary Line 10): Revise the description of the sedimentation height.

Response: We appreciate the reviewer's insightful comments. In response, we have added statistical analysis and included a statistical statement in the figure caption. To maintain clarity and avoid overcrowding the figure, we chose not to add asterisks directly to the image. Additionally, we revised the figure legend to address the reviewer's suggestions on repeating units and the description of sedimentation heights. The updated caption for Supplementary Figure 2 (previously Supplementary Figure 1) now reads:

“Supplementary Figure 2. Sedimentation was accelerated by adding polystyrene (PS) microplastics. (a) sedimentation without addition of polystyrene microplastics. (b) sedimentation with 0.05% (w/v) polystyrene microplastics. Time points from left to right are 0, 10, 20, 30, 40, 50, and 60 min after sedimentation. (c) quantification of sedimentation with /without polystyrene. To determine the height of sedimentation, cyanobacterial cells with /without polystyrene were allowed to settle in a graduated cylinder, and the height was measured by recording the position of the upper edge of the dark green sediment layer against the cylinder's scale. Significant differences ($p < 0.05$) were observed at time points of 10, 20, 30, 40, and 50 mins. Source data is provided as a Source Data file.”

Comment 35: (Supplementary Line 15-20): Revise and clarify the confusing text.

Response: We appreciate the reviewer pointing out that the legend for Supplementary Figure 3 (originally Supplementary Figure 2). We have revised the section and now the figure legend reads (Supplementary Line 51-61):

“Supplementary Figure 3. Evaluation of the separation of PS microplastics and cyanobacterial cells using low-speed centrifugation. The mixture of cyanobacterial cells and 200 nm polystyrene microplastics was separated by centrifugation at 800 ×g for 3 minutes. Under these conditions, nearly all cyanobacterial cells (with only about 1.5% of WT and HCC cells remaining in suspension) were pelleted, while the majority of 200 nm polystyrene microplastics (97.9%, $p < 0.01$) remained suspended. For mixtures containing 500 nm and 800 nm polystyrene microplastics, centrifugation was performed at 300 ×g for 3 minutes. In this case, 5.2% of WT cells and 2.0% of HCC cells remained in suspension, whereas most of the polystyrene microplastics - 90.4% of the 500 nm and 82.3% of the 800 nm particles - remained suspended ($p < 0.01$). These results demonstrate that low-speed centrifugation is an effective method for separating and estimating the content of cell-

microplastic mixtures. Data are presented as mean values \pm standard deviations (n = 3). Source data are provided as a Source Data file.”

Comment 36: (Supplementary Line 27-29): Based on the images provided, the results appear to suggest the opposite. Revise.

Response: We appreciate the reviewer pointing out the confusion regarding the figure legend. We have thoroughly revised this section, and the new figure legend now reads (Supplementary Line 65-70): *“Supplementary Figure 4. Solid content visualization of suspensions in the WT and HCC-treated samples. After microplastic removal by sedimentation, the suspension samples (upper layer as indicated in the figure) were centrifuged to visualize the solid contents in the suspension samples. The solid contents after centrifugation were higher in the WT samples than those in the HCC samples. Additionally, HCC samples with higher polystyrene concentrations exhibited lower solid content compared to those with lower concentrations. Additionally, lower solid content was observed in HCC samples with higher polystyrene concentration than those with lower concentrations, indicating the microplastic capture capacity of the engineered cyanobacteria. The experiment was repeated three times and similar results were obtained.”*

Comment 37: (Supplementary Figure 4): Revise the figure legend. Expressions like “Substantial higher microplastic removal rate” lack scientific meaning and should be replaced with statistically supported analyses. The statement “b. sedimentation was observed in the HCC samples but not in the WT samples” is not evident from the provided images. It is recommended to review the images, comparing controls and PS samples. Ensure that sample nomenclature is consistent throughout the figure and text.

Response: We sincerely appreciate the reviewer’s thoughtful comments and suggestions. In response, we have revised the figure caption to include a detailed description of the statistical analysis performed. Due to the addition of new data and figures, the original supplementary Figure 4 now becomes supplementary Figure 5. We have also updated the figure to ensure that the nomenclature is consistent throughout. To address the reviewer’s point regarding the comparison between controls and PS samples, we have added the statement “regardless of the presence of microplastics” to acknowledge this important aspect. However, it is challenging to visually distinguish differences between control and PS samples, and these quantitative comparative results are presented in Supplementary Figure 5a (attached below).

Regarding the comment on the statement that “sedimentation was observed in the HCC samples but not in the WT samples” and the concern about insufficient supporting evidence, we believe there may have been some misunderstanding due to an unclear figure legend. We have revised the legend to clarify this point. As shown in Supplementary Figure 5b (attached below), there is a clear difference in sedimentation between WT and HCC samples, both in the presence and absence of PS.

The revised Supplementary Figure 5 legend now reads: “**Supplementary Figure 5. Microplastic removal tests for 500 nm and 800 nm polystyrene microplastics.** *a. Significantly ($p < 0.01$) higher microplastic removal rate was observed in the HCC samples compared to the WT samples. The suspension turbidity after low-speed centrifugation of WT samples for each treatment was normalized to 100%. The turbidity of the centrifuged suspension in the HCC samples accounted for only 0.6% and 1.8% of that in the WT samples when 0.02% of 500 nm and 800 nm polystyrene microplastics were applied, respectively. These values slightly increased to 3.3% and 2.5% when 0.1% of 500 nm and 800 nm polystyrene microplastics were applied, respectively.* *b. Sedimentation was observed in the HCC samples, but not in the WT samples, regardless of the presence of microplastics, as indicated by the accumulation of sediments at the bottom of the sample and the lighter color of the suspension. The experiment was repeated three times and similar results were obtained. ** indicates $p < 0.01$.*”

Comment 38: (Supplementary Figure 6): The figure legend needs to be completed, providing the exact meaning of symbols such as **, +, and -, as well as the corresponding significance levels.

Response: We appreciate the reviewer’s suggestions. We have revised the Figure legend for Supplementary Figure 7 (originally Supplementary Figure 6) and it now reads: **“Supplementary Figure 7. Surfactant blocking to verify the hydrophobic interactions. Tween 20, a surfactant consisting of a hydrophilic head and a hydrophobic tail, was used to block the hydrophobic surfaces of both cyanobacteria and microplastics (a). When added to the samples, the hydrophobic tails of Tween 20 molecules interacted with the hydrophobic interfaces of the cyanobacterial cells and microplastics (MP), exposing their hydrophilic heads to the solution (a). This was expected to inhibit cell-to-cell and cell-to-microplastic aggregations, which was supported by the significantly higher suspension turbidities observed when Tween 20 was added to the samples (b). Further analysis of polystyrene abundance in suspension (after low-speed centrifugation) also validated that Tween 20 substantially blocked microplastic removal. These results suggest that the hydrophobicity effect is the driving force for cell sedimentation and microplastic removal. Data are presented as mean values \pm standard deviations ($n = 3$). ** indicate $p < 0.01$. The symbols “+” and “-” indicate the presence or absence, respectively, of the specified components in the treatments. Source data is provided as a Source Data file.”**

Comment 39: (Supplementary Figure 8): Specify the number of cultivation days to which the results refer. Additionally, statistical analysis between series and concentrations must be included to ensure the observed differences are adequately justified. In the phrase “...resulting in less contact with the cyanobacterial cells,” consider whether reduced shading caused by the microplastics is the main factor. A detailed discussion of this possibility is recommended.

Response: We appreciate the reviewer’s insightful comments. We have revised the figure and the figure legend to clarify the cultivation days and describe the details of the statistics. The content now reads: **“Supplementary Figure 9. Evaluation of cyanobacteria growth inhibition by microplastics. Polystyrene (PS) microplastics with diameters of 200 nm and 800 nm, as well as PET microplastics ($<300 \mu\text{m}$), were each added to cyanobacterial cells during cultivation. The cells were cultivated under illumination of $150 \mu\text{mol}/\text{m}^2/\text{s}$ at a temperature of $39 \text{ }^\circ\text{C}$, with a shaking speed of 400 rpm, in a CO_2 chamber set to 1% CO_2 for 2 days. The growth of cyanobacteria was estimated using chlorophyll fluorescence (Ex 595 nm and Em 690 nm). The chlorophyll fluorescence of the control group (without microplastic addition) was normalized to 100%. The results showed that at low concentrations, microplastics had marginal impacts on cyanobacterial growth. As the concentration increased, both 200 nm and 800 nm microplastics negatively affected cyanobacterial growth, with the smaller diameter microplastics having a stronger inhibitory effect. No**

significant growth inhibition was observed at any concentration tested for PET microplastics, presumably due to their larger size, which causes them to settle at the bottom of the growth container, resulting in less contact with the cyanobacterial cells and reduced shading on cyanobacterial cells. Data are presented as mean values \pm standard deviations ($n = 3$ independent samples with three technical replicates). * indicates $p < 0.05$. ** indicates $p < 0.01$. Source data is provided as a Source Data file.”

Comment 40: (Supplementary Figure 10): Provide statistical analysis and review the sample nomenclature.

Response: We appreciate the reviewer’s comments. We have revised the figure and legend. As such, “PS”, “WT-PS”, and “HCC-PS” were used to be consistent with the entire manuscript. The Supplementary Figure 10b (originally Supplementary Figure 10) and its legend are included below:

“(b) Modulus of elasticity comparison. No significant differences ($p > 0.05$) were observed between the polystyrene (PS) plastics and bioplastics made from WT-PS or HCC-PS samples, in terms of modulus of elasticity. Data is presented as mean values \pm standard deviations ($n > 5$).”

In summary, we would also like to thank the reviewers again for your acknowledgement of the potential of our work. Your instructive suggestions provided directions for improvement. We have performed more experiments, conducted more analyses, and carefully edited our manuscript based on the feedback. With the new data and revision, we hope that we have addressed the concerns raised in the review process and that our manuscript meets the publication standards for Nature Communications.

Reference

- Abdelfattah, A., Ali, S.S., Ramadan, H., El-Aswar, E.I., Eltawab, R., Ho, S.H., Elsamahy, T., Li, S., El-Sheekh, M.M., Schagerl, M., Kornaros, M. and Sun, J. 2023. Microalgae-based wastewater treatment: Mechanisms, challenges, recent advances, and future prospects. *Environ Sci Ecotechnol* 13, 100205.
- Campan, M., Nihart, A., Garcia, M., Liu, R., Olewine, M., Castillo, E., Bleske, B., Scott, J., Howard, T., Gonzalez-Estrella, J., Adolphi, N., Gallego, D. and Hayek, E.E. 2024. Bioaccumulation of Microplastics in Decedent Human Brains Assessed by Pyrolysis Gas Chromatography-Mass Spectrometry. PREPRINT (Version 1) available at Research Square
- Cholewinski, A., Dadzie, E., Sherlock, C., Anderson, W.A., Charles, T.C., Habib, K., Young, S.B. and Zhao, B. 2022. A critical review of microplastic degradation and material flow analysis towards a circular economy. *Environ Pollut* 315, 120334.
- Chu, C.H., Lin, Y.X., Liu, C.K. and Lai, M.C. 2023. Development of Innovative Online Modularized Device for Turbidity Monitoring. *Sensors (Basel)* 23(6).
- Hammond, C.R., Hernandez, M.S.G. and Loge, F.J. 2025. Microalgal-bacterial aggregates for wastewater treatment: Origins, challenges, and future directions. *Water Environ Res* 97(2), e70018.
- Lee, H., Xu, V., Diao, J., Zhao, R., Chen, M., Moon, T.S., Liu, H., Parker, K.M., Jun, Y.-S. and Tang, Y.J. 2024. The use of a benign fast-growing cyanobacterial species to control microcystin synthesis from *Microcystis aeruginosa*. *Frontiers in Microbiology* Volume 15 - 2024.
- Liu, Z.H., Hao, N., Wang, Y.Y., Dou, C., Lin, F., Shen, R., Bura, R., Hodge, D.B., Dale, B.E., Ragauskas, A.J., Yang, B. and Yuan, J.S. 2021. Transforming biorefinery designs with 'Plug-In Processes of Lignin' to enable economic waste valorization. *Nat Commun* 12(1), 3912.
- Long, B., Fischer, B., Zeng, Y.N., Amerigian, Z., Li, Q., Bryant, H., Li, M., Dai, S.Y. and Yuan, J.S. 2022. Machine learning-informed and synthetic biology-enabled semi-continuous algal cultivation to unleash renewable fuel productivity. *Nat Commun* 13(1).
- Reddy, A.S. and Nair, A.T. 2022. The fate of microplastics in wastewater treatment plants: An overview of source and remediation technologies. *Environ Technol Inno* 28.
- Roh, H., Lee, J.S., Choi, H.I., Sung, Y.J., Choi, S.Y., Woo, H.M. and Sim, S.J. 2021. Improved CO₂-derived polyhydroxybutyrate (PHB) production by engineering fast-growing cyanobacterium *Synechococcus elongatus* UTEX 2973 for potential utilization of flue gas. *Bioresource Technology* 327, 124789.
- Rosenberg, M., Gutnick, D. and Rosenberg, E. 1980. Adherence of bacteria to hydrocarbons: a simple method for measuring cell-surface hydrophobicity. *FEMS Microbiol lett* 9(1), 29-33.
- Samiotis, G., Stamatakis, K. and Amanatidou, E. 2021. Assessment of *Synechococcus elongatus* PCC 7942 as an option for sustainable wastewater treatment. *Water Science and Technology* 84(6), 1438-1451.

- Samiotis, G., Stamatakis, K. and Amanatidou, E. 2022. Dimensioning of *Synechococcus elongatus* PCC 7492 cultivation photobioreactor for valorization of wastewater resources. *Chem Eng J* 435, 134895.
- Sarcletti, M., Park, H., Wirth, J., Englisch, S., Eigen, A., Drobek, D., Vivod, D., Friedrich, B., Tietze, R., Alexiou, C., Zahn, D., Zubiri, B.A., Spiecker, E. and Halik, M. 2021. The remediation of nano-/microplastics from water. *Mater Today* 48, 38-46.
- Shen, M., Zhang, Y., Almatrafi, E., Hu, T., Zhou, C., Song, B., Zeng, Z. and Zeng, G. 2022. Efficient removal of microplastics from wastewater by an electrocoagulation process. *Chem Eng J* 428, 131161.
- Su, Y., Mennerich, A. and Urban, B. 2011. Municipal wastewater treatment and biomass accumulation with a wastewater-born and settleable algal-bacterial culture. *Water Res* 45(11), 3351-3358.
- Wang, H., Neal, B., White, B., Nelson, B., Lai, J., Long, B., Arreola-Vargas, J., Yu, J., Banik, M.T. and Dai, S.Y. 2023. Microplastics removal in the aquatic environment via fungal pelletization. *Bioresour Tech Rep* 23, 101545.
- Yu, J., Liberton, M., Cliften, P.F., Head, R.D., Jacobs, J.M., Smith, R.D., Koppenaar, D.W., Brand, J.J. and Pakrasi, H.B. 2015. *Synechococcus elongatus* UTEX 2973, a fast growing cyanobacterial chassis for biosynthesis using light and CO₂. *Sci Rep-Uk* 5(1), 8132.
- Zhang, L., Chen, L., Diao, J., Song, X., Shi, M. and Zhang, W. 2020. Construction and analysis of an artificial consortium based on the fast-growing cyanobacterium *Synechococcus elongatus* UTEX 2973 to produce the platform chemical 3-hydroxypropionic acid from CO₂. *Biotechnology for Biofuels* 13(1), 82.
- Zhang, Y., Whalen, J.K., Cai, C., Shan, K. and Zhou, H. 2023. Harmful cyanobacteria-diatom/dinoflagellate blooms and their cyanotoxins in freshwaters: A nonnegligible chronic health and ecological hazard. *Water Research* 233, 119807.

Response to Reviewer Comments

Response: We sincerely thank all four reviewers for their thoughtful and constructive comments, which have been invaluable in improving the quality and clarity of our work. We are glad the first three reviewers are satisfied with our responses. In this second round of response, we have carefully revised both the manuscript and the Supplementary Information to address each point raised. The specific changes made are detailed below, and we hope these revisions meet the reviewer's expectations.

Reviewer 4

The environmental impacts presented in Supplementary Table 7 are evaluated using the CML v4.0 (2016) methodology, and each impact category is reported with its corresponding functional unit. The inclusion of avoided burden credits (e.g., for electricity and plastic film displacement) is a reasonable approach for system expansion. However, the following clarifications should be provided:

(a) Details (life cycle emission factor for the displaced conventional plastic in kg CO₂-eq/kg conventional plastic) on what is displaced (e.g., grid mix, plastic).

(b) The value of how much bioplastic displaces conventional plastic on a functional equivalence basis.

(Table S2 and Figure S11) Supplementary Table 2 provides a useful summary of the primary process inputs and outputs per liter of wastewater.

The system boundary includes "algae storage" but does not quantify or discuss potential emissions from this stage (e.g., biomass degradation, VOC, or CH₄/N₂O emissions during wet storage). The authors are encouraged to either clarify that storage emissions are assumed negligible or to justify this simplification with supporting references or estimates.

There are also concerns regarding the parameter assumptions used in the techno-economic analysis, specifically:

(a) Operating labor costs are not explicitly reported in Supplementary Table 4. While the fixed capital investment typically includes installation labor, the lack of clarity regarding operating labor cost estimation may lead to an underestimation of annual operating expenses.

(b) The authors should clarify how indirect equipment costs (e.g., installation, instrumentation, piping) were estimated. Is it based on empirical parametric estimation/database/ tools or software to perform the TEA modeling? A recent article (DOI: 10.1016/j.cep.2024.110001) highlights how neglecting installation and labor-related indirect costs can lead to optimistic cost projections. I suggest the authors provide more details to ensure reproducibility of their economic assessment.

Response: We sincerely thank the reviewers for their thoughtful and constructive comments, which have been very helpful in improving the quality and clarity of our work. In response, we have carefully revised both the manuscript and the Supplementary Information to address each point raised. We have quoted all major revisions in the main text and supporting information (*all in italic and underlined*), with the revised parts highlighted in red. The specific changes made are detailed below, and we hope these revisions meet the reviewer's expectations.

1. 1. Clarification on displaced products and emission factors

(a) Life cycle emission factor for displaced conventional plastic:

Thank you for this suggestion. We have added the life cycle GHG emission factor for displaced conventional plastic film in the note of Supplementary Table 2 and clarified the data sources. The note now reads: **Note: Untreated wastewater is used as the baseline for comparison.** ** With optimized strain selection, metabolic engineering, and cultivation under stress, microalgal systems could potentially reach ~50–70 wt%, although this remains largely unachieved in practice. **The electricity factor is 0.44368 kg CO₂-eq per kWh, also from Ecoinvent 3.9, for production from a natural gas combined-cycle power plant in Texas, USA (UUID: 8c40b198-fdcf-4ded-9def-ec647470107d). ***The emission factor is 0.65167 kg CO₂-eq per kg of plastic film (UUID: f5c54f00-b673-4a59-9df5-ef88a970cb20) from Ecoinvent 3.9. Assume a 1:1 functional equivalence by bioplastic mass which is applicable to products such as agricultural mulch films, landscaping covers, construction sheets, and other applications^{1,2}.* in L190 - L196 in supplementary information.

(b) Extent of bioplastic displacement on a functional equivalence basis:

We have added the life cycle GHG emission factor for displaced conventional plastic film in the note of Supplementary Table 2 that displacement is calculated on a 1:1 functional equivalence by mass, assuming that 1 kg of our bioplastic replaces 1 kg of conventional plastic products, include agricultural mulch films, landscaping covers, construction sheets, and other applications². This assumption is justified by the material's performance profile. While the bioplastic's tensile strength is lower than that of PS (17.0 vs. 25.6 MPa), its toughness is substantially higher (545.3 vs. 244.2 J/m²) and its elongation is more than doubled (4.2% vs. 1.8%). This enhanced durability directly counters the primary failure mode of PS in target applications like rigid containers—brittle fracture. As the material can fulfill its core function without requiring additional mass, the 1:1 displacement ratio is a reasonable and conservative basis for this analysis. The revised Note now reads: **Note: Untreated wastewater is used as the baseline for comparison.** ** With optimized strain selection, metabolic engineering, and cultivation under stress, microalgal systems could potentially reach ~50–70 wt%, although this remains largely unachieved in practice. **The electricity factor is 0.44368 kg CO₂-eq per kWh, also from Ecoinvent 3.9, for production from a natural gas combined-cycle power plant in Texas, USA (Universally Unique Identifier in the Ecoinvent database (UUID): 8c40b198-fdcf-4ded-9def-ec647470107d). ***The emission factor is 0.65167 kg CO₂-eq per kg of plastic film (UUID): f5c54f00-b673-4a59-9df5-*

ef88a970cb20) from Ecoinvent 3.9. Assume a 1:1 functional equivalence by bioplastic mass which is applicable to products such as agricultural mulch films, landscaping covers, construction sheets, and other applications^{1,2}.” in L190 - L196 in supplementary information.

2. Algae storage emissions

We have clarified in the caption of Supplementary Figure 11 that emissions from the algae storage stage are assumed to be negligible. Our justification is discussed below:

As our storage purpose is to prepare the next step of algae biomass usage, the storage conditions will favor the least biomass decay and minimum emissions. Thus, assuming negligible emissions during storage is reasonable and consistent with practices where the algae biomass is intended for future usage^{3,4}. Indeed, our system storage stage involves only small quantities of biomass intended to buffer or support continuous open-pond or photobioreactor cultivation, minimizing any potential environmental impacts from this step relative to the total life cycle. Based on published estimates from tropical algal facultative ponds⁵, average greenhouse gas fluxes are approximately 2.36 g CH₄/m²·d, 1.75 g CO₂/m²·d, and 0.00285 g N₂O/m²·d. Normalizing these values to our system yield of 5.3 kg algae per m² gives corresponding emissions of 0.445 g CH₄, 0.330 g CO₂, and 0.00054 g N₂O per kilogram of algae per day. Converting to CO₂ equivalents (100-year GWPs: CH₄ = 28, CO₂ = 1, N₂O = 265), the total is approximately 0.013 g CO₂-eq per gram of algae, with around 90 % of the climate impact arising from methane. Literature on anaerobic ensiling of microalgae⁶ indicates that only 5–10 % dry matter loss occurs in the first month, with emissions dominated by CO₂ and trace fermentation gases and minimal CH₄ due to rapid acidification—a process that suggests at least an 80–90 % reduction in CO₂-eq emissions compared to open ponds. Applying this reduction, the three-day buffer storage of seed algae associated with treating 1 L of wastewater would generate only ~0.0006 g CO₂-eq—and likely even less given the very short storage duration—making its contribution effectively negligible in the overall greenhouse gas balance. The revised caption for Supplementary Figure 11 now reads: “**Supplementary Figure 11. The boundary of the RUMBA-based biomass production, wastewater treatment, microplastic removal and upcycling system. The background data are referenced from Ecoinvent 3.9, which includes the upstream manufacturing processes, chemical, electricity, and wastewater treatment inventory data. The foreground system comprises algae cultivation, biomass recovery, and biomass separation for bioplastic production, as determined by experimental data. The short-term algae storage stage is assumed to be negligible, consistent with the practice 1) general LCA analysis on algal biofuel application system does not take account of CO₂ emission^{3,4,7}, 2) an emission estimate of an open pond system that operates for months long durations^{5,6}. Our algal storage is intended for about three days, which resulted in about 0.0006g CO₂-eq/L wastewater, which is negligible to the total CO₂ emission. For system expansion, we assumed that the residual biomass can be used for electricity generation and sold to the grid.**” in L136 – L146 in supplementary information.

3. Techno-economic analysis (TEA) parameter clarifications

(a) Operating labor costs:

We have now explicitly reported the annual operating labor costs in Supplementary Table 6. Operating labor was estimated based on staffing requirements for each major process area and regional wage rates⁸. Note that our plant is integrated with the entire wastewater treatment facility,

so we assume to share labor with it, allowing for a labor allocation of 0.5. To reflect this addition, we have revised the sentence in L439–L440 to read: “*Economic assumptions, equipment costs, labor costs, and capital costs are detailed in Supplementary Tables 3-7.*” now including a citation to Supplementary Table 6. The newly added table is provided below.

Supplementary Table 6. Labor Cost Breakdown

Position	2024 Salary (\$)
Plant Manager*	77,809
Plant Engineer (civil) *	41,025
Plant Engineer (environmental) *	41,680
Maintenance Tech	42,345
Lab Manager*	29,641
Lab Technician	42,345
Shift Supervisor	50,814
Module operator - Production	26,910
Module operator - Dewatering	38,590
Clerks & Secretaries	38,110
Total Salaries	410,214
Labor Burden (90%)	369,193

Note: * Labor requirements are assumed to have a 50% allocation through integration with the wastewater treatment facility, as duties are shared between the wastewater treatment and algae operations.

(b) Indirect equipment cost estimation:

We have added details in **Supplementary Table 7** specifying that total capital costs include both direct purchased equipment costs and indirect costs. The total installed cost was estimated using a Lang factor of 3, applied to the sum of purchased equipment, piping and control systems, and balance-of-plant costs. These factors are derived from historical chemical process plant data and adapted for algae/bioprocess systems^{9,10}. No commercial software was used; instead, we relied on engineering cost correlations and literature precedents. We also reference the recent work (DOI: 10.1016/j.cep.2024.110001) and note that our inclusion of installation and labor-related indirect costs helps avoid the optimistic bias described in that study. To incorporate this addition, we have updated the sentence in L439–L440 to state: “*Economic assumptions, equipment costs, labor costs, and capital costs are detailed in Supplementary Tables 3-7.*” now referencing the newly included Supplementary Table 7. The newly added Supplementary Table 7 is shown below:

Supplementary Table 7: Total Capital Cost

Open Pond System (\$)	
Storage	62,532
Open pond	75,675
Separation and dryer	127,998
Extraction	681,630
Piping & Control system	88,530
Balance of Plant	310,910
Total purchased cost	1,036,365
Lang factor	3.0
Total capital cost	4,041,824
PBR System (\$)	
Storage	62,532
Open pond	15,955,515
Separation and dryer	2,685,760
Extraction	681,630
Piping & Control system	1,932,291
Balance of Plant	6,395,318
Total purchased cost	21,317,728
Lang factor	3.0
Total capital cost	83,139,138

References

1. Heimersson, S., Morgan-Sagastume, F., Peters, G. M., Werker, A. & Svanström, M. Methodological issues in life cycle assessment of mixed-culture polyhydroxyalkanoate production utilising waste as feedstock. *New Biotechnology* **31**, 383–393 (2014).
2. Nizamuddin, S., Baloch, A. J., Chen, C., Arif, M. & Mubarak, N. M. Bio-based plastics, biodegradable plastics, and compostable plastics: biodegradation mechanism, biodegradability standards and environmental stratagem. *International Biodeterioration & Biodegradation* **195**, 105887 (2024).
3. Ou, L. *et al.* Utilizing high-purity carbon dioxide sources for algae cultivation and biofuel production in the United States: Opportunities and challenges. *Journal of Cleaner Production* **321**, 128779 (2021).
4. Singh, U., Banerjee, S. & Hawkins, T. R. Implications of CO₂ Sourcing on the Life-Cycle Greenhouse Gas Emissions and Costs of Algae Biofuels. *ACS Sustainable Chem. Eng.* **11**, 14435–14444 (2023).
5. Vinasco, J. P. S. GHG emissions from algal facultative ponds under tropical conditions. in *Greenhouse Gas Emissions from Ecotechnologies for Wastewater Treatment* (CRC Press, 2021).
6. Babich, O. *et al.* Fermentation of micro- and macroalgae as a way to produce value-added products. *Biotechnology Reports* **41**, e00827 (2024).
7. Álvarez-González, A. *et al.* Environmental and economic benefits of using microalgae grown in wastewater as biofertilizer for lettuce cultivation. *Bioresource Technology* **424**, 132230 (2025).
8. Davis, R. *et al.* *Process Design and Economics for the Conversion of Lignocellulosic Biomass to Hydrocarbons: Dilute-Acid and Enzymatic Deconstruction of Biomass to Sugars and*

Biological Conversion of Sugars to Hydrocarbons. <https://www.osti.gov/biblio/1107470>

(2013) doi:10.2172/1107470.

9. Peters, M. S., Timmerhaus, K. D. & West, R. E. *Plant Design and Economics for Chemical Engineers*. (McGraw-Hill Education, 2003).

10. Klein, B. & Davis, R. *Algal Biomass Production via Open Pond Algae Farm Cultivation: 2021 State of Technology and Future Research*. <https://www.osti.gov/biblio/1862662> (2022)
doi:10.2172/1862662.

Response to Reviewer Comments

Reviewer 4

The authors have addressed the reviewer's comments and improved the quality of TEA and LCA in their manuscript.

Response: We sincerely appreciate the reviewer's thoughtful comments and constructive suggestions throughout the review process. We are grateful that the reviewer recognizes the improvements in the TEA and LCA.